# Lapses in perceptual decisions reflect exploration

**Sashank Pisupati[1,2†], Lital Chartarifsky-Lynn[1,2†], Anup Khanal[1], Anne K Churchland[3]***

[1]Cold Spring Harbor Laboratory, Cold Spring Harbor, New York, United States;
[2]CSHL School of Biological Sciences, Cold Spring Harbor, New York, United States;
[3]University of California, Los Angeles, Los Angeles, United States

**Abstract** Perceptual decision-makers often display a constant rate of errors independent of evidence strength. These 'lapses' are treated as a nuisance arising from noise tangential to the decision, e.g. inattention or motor errors. Here, we use a multisensory decision task in rats to demonstrate that these explanations cannot account for lapses' stimulus dependence. We propose a novel explanation: lapses reflect a strategic trade-off between exploiting known rewarding actions and exploring uncertain ones. We tested this model's predictions by selectively manipulating one action's reward magnitude or probability. As uniquely predicted by this model, changes were restricted to lapses associated with that action. Finally, we show that lapses are a powerful tool for assigning decision-related computations to neural structures based on disruption experiments (here, posterior striatum and secondary motor cortex). These results suggest that lapses reflect an integral component of decision-making and are informative about action values in normal and disrupted brain states.

**\*For correspondence:**
AChurchland@mednet.ucla.edu

[†]These authors contributed equally to this work

**Competing interests:** The authors declare that no competing interests exist.

## Introduction

Perceptual decisions are often modeled using noisy ideal observers (e.g., Signal detection theory, *Green and Swets, 1966*; Bayesian decision theory, *Dayan and Daw, 2008*) that explain subjects' errors as a consequence of noise in sensory evidence. This predicts an error rate that decreases with increasing sensory evidence, capturing the sigmoidal relationship often seen between evidence strength and subjects' decision probabilities (i.e. the psychometric function).

Human and nonhuman subjects often deviate from these predictions, displaying an additional constant rate of errors independent of the evidence strength known as 'lapses', leading to errors even on extreme stimulus levels (*Wichmann and Hill, 2001*; *Busse et al., 2011*; *Gold and Ding, 2013*; *Carandini and Churchland, 2013*). Despite the knowledge that ignoring or improperly fitting lapses can lead to serious mis-estimation of psychometric parameters (*Wichmann and Hill, 2001*; *Prins and Kingdom, 2018*), the cognitive mechanisms underlying lapses remain poorly understood. A number of possible sources of noise have been proposed to explain lapses, typically tangential to the decision-making process.

One class of explanations for lapses relies on pre-decision noise added due to fluctuating attention, which is often operationalized as a small fraction of trials on which the subject fails to attend to the stimulus (*Wichmann and Hill, 2001*). On these trials, it is assumed that the subject cannot specify the stimulus (i.e. sensory noise with infinite variance, *Bays et al., 2009*) and hence guesses randomly or in proportion to prior beliefs. This model can be thought of as a limiting case of the Variable Precision model, which assumes that fluctuating attention has a more graded effect of scaling the sensory noise variance (*Garrido et al., 2011*), giving rise to heavy tailed estimate distributions, resembling lapses in the limit of high variability (*Shen and Ma, 2019*; *Zhou et al., 2018*). Temporal

forms of inattention have also been proposed to give rise to lapses, where the animal ignores early or late parts of the evidence (impulsive or leaky integration, *Erlich et al., 2015*).

An alternative class of explanations for lapses relies on a fixed amount of noise added after a decision has been made, commonly referred to as 'post-categorization' noise (*Erlich et al., 2015*) or decision noise (*Law and Gold, 2009*). Such noise could arise from errors in motor execution (e.g. finger errors, *Wichmann and Hill, 2001*), non-stationarities in the decision rule arising from computational imprecision (*Findling et al., 2018*), suboptimal weighting of choice or outcome history (*Roy et al., 2018*; *Busse et al., 2011*), or random variability added for the purpose of exploration (e. g. 'ε-greedy' decision rules).

A number of recent observations have cast doubt on fixed early- or late-stage noise as satisfactory explanations for lapses. For instance, many of these explanations predict that lapses should occur at a constant rate, while in reality, lapses are known to reduce in frequency with learning in nonhuman primates (*Law and Gold, 2009*; *Cloherty et al., 2019*). Further, they can occur with different frequencies for different stimuli even within the same subject (in rodents, *Nikbakht et al., 2018*; and humans, *Mihali et al., 2018*; *Bertolini et al., 2015*; *Flesch et al., 2018*), suggesting that they may reflect task-specific, associative processes that can vary within a subject.

Lapse frequencies are even more variable across subjects and can depend on the subject's age and state of brain function. For instance, lapses are significantly higher in children and patient populations than in healthy adult humans (*Roach et al., 2004*; *Witton et al., 2017*; *Manning et al., 2018*). Moreover, a number of recent studies in rodents have found that perturbing neural activity in secondary motor cortex (*Erlich et al., 2015*) and striatum (*Yartsev et al., 2018*; *Guo et al., 2018*) has dramatic, asymmetric effects on lapses in auditory decision-making tasks. Because these perturbations were made in structures known to be involved in action selection, an intriguing possibility is that lapses reflect an integral part of the decision-making process, rather than a peripheral source of noise. However, because these studies only tested auditory stimuli, they did not afford the opportunity to distinguish sensory modality-specific deficits from general decision-related deficits. Taken together, these observations point to the need for a deeper understanding of lapses that accounts for effects of stimulus set, learning, age, and neural perturbations.

Here, we leverage a multisensory decision-making task in rodents to reveal the inadequacy of traditional models. We challenge a key assumption of perceptual decision-making theories, i.e. subjects' perfect knowledge of expected rewards (*Dayan and Daw, 2008*), to uncover a novel explanation for lapses: uncertainty-guided exploration, a well-known strategy for balancing exploration and exploitation in value-based decisions. We test the predictions of the exploration model for perceptual decisions by manipulating the magnitude and probability of reward under conditions of varying uncertainty. Finally, we demonstrate that suppressing secondary motor cortex or posterior striatum unilaterally has an asymmetric effect on lapses that generalizes across sensory modalities, but only in uncertain conditions. This can be accounted for by an action value deficit contralateral to the inactivated side, reconciling the proposed perceptual and value-related roles of these areas and suggesting that lapses are informative about the subjective values of actions, reflecting a core component of decision-making.

## Results

### Testing ideal observer predictions in perceptual decision-making

We leveraged an established decision-making task (*Raposo et al., 2012*; *Sheppard et al., 2013*; *Licata et al., 2017*) in which freely moving rats judge whether the fluctuating rate of a 1000 ms series of auditory clicks and/or visual flashes (rate range: 9–16 Hz) is high or low compared with an abstract category boundary of 12.5 Hz (*Figure 1a–c*). Using Bayesian decision theory, we constructed an ideal observer for our task that selects choices that maximize expected reward (see Materials and methods: Modeling). To test whether behavior matches ideal observer predictions, we presented multisensory trials with matched visual and auditory rates (i.e., both modalities carried the same number of events per second; *Figure 1c*, bottom) interleaved with visual-only or auditory-only trials. This allowed us to separately estimate the sensory noise in the animal's visual and auditory system, and compare the measured performance on multisensory trials to the predictions of the ideal observer.

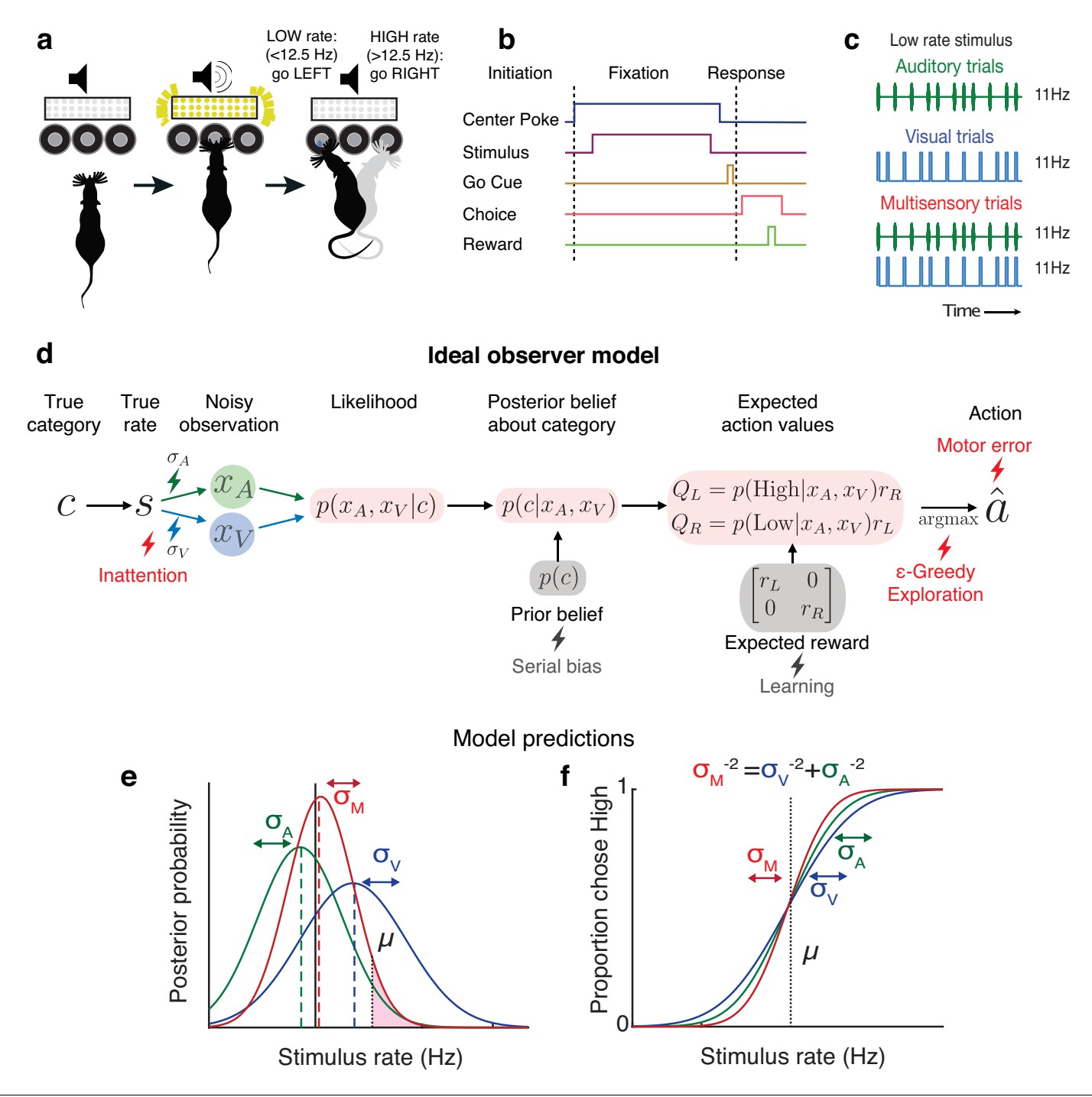

**Figure 1.** Testing ideal observer predictions in perceptual decision-making. (**a**) Schematic drawing of rate discrimination task. Rats initiate trials by poking into a center port. Trials consist of visual stimuli presented via a panel of diffused LEDs, auditory stimuli presented via a centrally positioned speaker, or multisensory stimuli presented from both. Rats are rewarded with a 24 μL drop of water for reporting high-rate stimuli (greater than 12.5 Hz) with rightward choices and low-rate stimuli (lower than 12.5 Hz) with leftward choices. (**b**) Timeline of task events. (**c**) Example stimulus on auditory (top), visual (middle), and multisensory trials (bottom). Stimuli consist of a stream of events separated by long (100 ms) or short (50 ms) intervals. Multisensory stimuli consist of visual and auditory streams carrying the same underlying rate. Visual, auditory, and multisensory trials were randomly interleaved (40% visual, 40% auditory, and 20% multisensory). (**d**) Schematic outlining the computations of a Bayesian ideal observer. Stimulus belonging to a true category $c$ with a true underlying rate $s$ gives rise to noisy observations $x_A$ and $x_V$, which are then integrated with each other and with prior beliefs to form a multisensory posterior belief about the category, and further combined with reward information to form expected action values $Q_L, Q_R$. The ideal observer selects the action $\hat{a}$ with maximum expected value. Lightning bolts denote proposed sources of noise that can give rise to (red) or exacerbate

*Figure 1 continued on next page*

*Figure 1 continued*

(gray) lapses, causing deviations from the ideal observer. (e) Posterior beliefs on an example trial assuming flat priors. Solid black line denotes true rate, and blue and green dotted lines denote noisy visual and auditory observations, with corresponding unisensory posteriors shown in solid blue and green. Solid red denotes the multisensory posterior, centered around the maximum a posteriori rate estimate in dotted red. Shaded fraction denotes the probability of the correct choice being rightward, with μ denoting the category boundary. (f) Ideal observer predictions for the psychometric curve, that is, proportion of high-rate choices for each rate. Inverse slopes of the curves in each condition are reflective of the posterior widths on those conditions, assuming flat priors. The value on the abscissa corresponding to the curve's midpoint indicates the subjective category boundary, assuming equal rewards and flat priors.

Performance was assessed using a psychometric curve, that is, the probability of high-rate decisions as a function of stimulus rate (*Figure 1f*). The ideal observer model predicts a relationship between the slope of the psychometric curve and noise in the animal's estimate: the higher the standard deviation (σ) of sensory noise, the more uncertain the animal's estimate of the rate and the shallower the psychometric curve. On multisensory trials, the ideal observer should have a more certain estimate of the rate (*Figure 1e*, visual [blue] and auditory [green] σ values are larger than multisensory σ [red]), driving a steeper psychometric curve (*Figure 1f*, red curve is steeper than green and blue curves). Since this model does not take lapses into account, it would predict perfect performance on the easiest stimuli on all conditions, and thus all curves should asymptote at 0 and 1 (*Figure 1f*).

## Lapses cause deviations from ideal observer and are reduced on multisensory trials

In practice, the shapes of empirically obtained psychometric curves do not perfectly match the ideal observer (*Figure 2*) since they asymptote at values that are less than 1 or greater than 0. This is a well-known phenomenon in psychophysics (*Wichmann and Hill, 2001*), requiring two additional lapse parameters to precisely capture the asymptotes. To account for lapses, we fit a four-parameter psychometric function to the subjects' choice data (*Figure 2a* – red, *Equation 1* in Materials and methods) with the Palamedes toolbox (*Prins and Kingdom, 2018*). γ and λ are the lower and upper asymptotes of the psychometric function, which parameterize lapses on low and high rates respectively; $\phi$ is a sigmoidal function, in our case the cumulative normal distribution; x is the event rate, that is, the average number of flashes or beeps presented during the 1 s stimulus period; μ parameterizes the midpoint of the psychometric function and σ describes the inverse slope after correcting for lapses.

How can we be sure that the asymptotes seen in the data truly reflect nonzero asymptotes rather than fitting artifacts or insufficient data at the asymptotes? To test whether lapses were truly necessary to explain the behavior, we fit the curves with and without lapses (*Figure 2b*) and tested whether the lapse parameters were warranted. The fit without lapses was rejected in 15/17 rats by the Bayes Information Criterion (BIC) and in all rats by the Akaike Information Criterion (AIC). Fitting a fixed lapse rate across conditions was not sufficient to capture the data, nor was fitting a lapse rate that was constrained to be less than 0.1 (*Wichmann and Hill, 2001*). Both data pooled across subjects and individual subject data warranted fitting separate lapse rates to each condition ('variable lapses' model outperforms 'fixed lapses', 'restricted lapses' or 'no lapses' in 13/17 individuals based on BIC, all individuals based on AIC and in pooled data based on both, *Figure 2g*).

Multisensory trials offer an additional, strong test of ideal observer predictions. In addition to perfect performance on the easiest stimuli, the ideal observer model predicts the minimum possible perceptual uncertainty achievable on multisensory trials through optimal integration (*Ernst and Bülthoff, 2004*; *Equation 9* in Materials and methods). By definition, better-than-optimal performance is impossible. However, studies in humans, rodents, and nonhuman primates performing multisensory decision-making tasks suggest that in practice, performance occasionally exceeds optimal predictions (*Raposo et al., 2012*; *Nikbakht et al., 2018*; *Hou et al., 2018*), seeming, at first, to violate the ideal observer model. Moreover, in these data sets, performance on the easiest stimuli was not perfect and asymptotes deviated from 0 and 1. As in these previous studies, when we fit performance without lapses, multisensory performance was significantly supra-optimal (p=0.0012, paired t-test), i.e. better than the ideal observer prediction (*Figure 2c*, black points are above the unity line). This was also true when lapse probabilities were assumed to be fixed across conditions

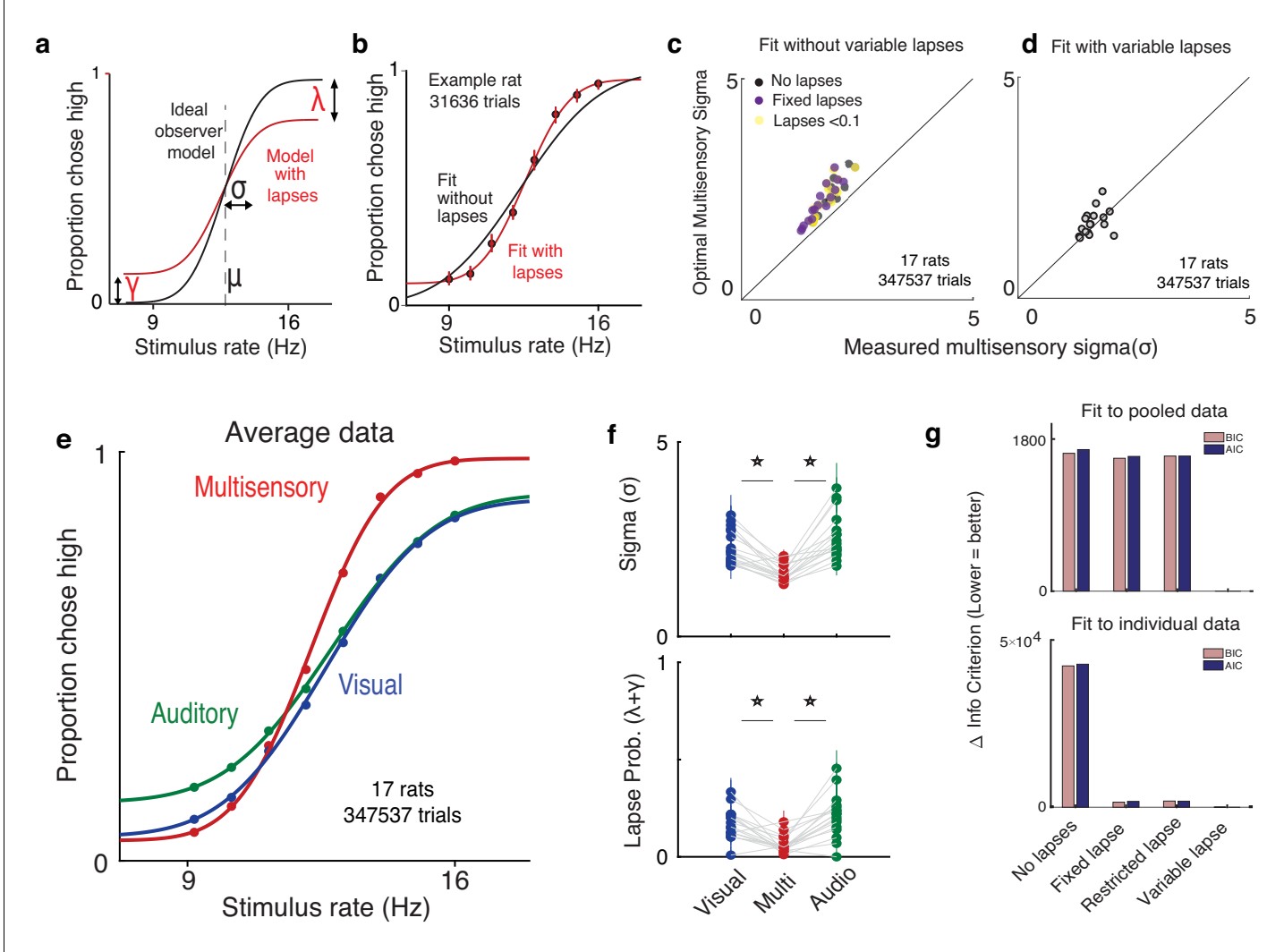

**Figure 2.** Deviations from ideal observer reflect lapses in judgment. (**a**) Schematic psychometric performance of an ideal observer (black) vs. a model that includes lapses (red). The ideal observer model includes two parameters: midpoint ($\mu$) and inverse slope ($\sigma$). The four-parameter model includes $\mu$, $\sigma$, and lapse probabilities for low-rate ($\gamma$) and high-rate choices ($\lambda$). Dotted line shows the true category boundary (12.5 Hz). (**b**) Subject data was fit with a two-parameter model without lapses (black) and a four-parameter model with lapses (red). (**c and d**) Ideal observer predictions vs. measured multisensory sigma for fits with and without variable lapses across conditions. (**c**) Multisensory integration seems supra-optimal if lapses are not accounted for (no lapses, black), fixed across conditions (fixed lapses, purple), or assumed to be less than 0.1 (restricted lapses, yellow). (**d**) Optimal multisensory integration is restored when allowing lapses to vary freely across conditions (n = 17 rats. Points represent individual rats. Data points that lie on the unity line represent cases in which the measured sigma was equal to the optimal prediction). (**e**) Rats' psychometric curves on auditory (green), visual (blue), and multisensory (red) trials. Points represent data pooled across 17 rats, and lines represent separate four-parameter fits to each condition. (**f**) Fit values of sigma (top) and lapse parameters (bottom) on unisensory and multisensory conditions. Both parameters showed significant reduction on the multisensory conditions (paired t-test, p<0.05); n = 17 rats (347,537 trials). (**g**) Model comparison using Bayes Information Criterion (pink) and Akaike Information Criterion (blue) for fits to pooled data across subjects (top) and to individual subject data (bottom). Lower scores indicate better fits. Both metrics favor a model where lapses are allowed to vary freely across conditions ('Variable lapse') over one without lapses ('No lapses'), one with a fixed probability of lapses ('Fixed lapse'), or where the lapses are restricted to being less than 0.1 ('Restricted lapse').

(p=0.0018, *Figure 2c*, purple) or when they were assumed to be less than 0.1 (p=0.0003, *Figure 2c*, yellow). However, when we allowed lapses to vary freely across conditions, performance was indistinguishable from optimal (*Figure 2d*, data points are on the unity line). This reaffirms that proper treatment of lapses is crucial for accurate estimation of perceptual parameters and offers a potential explanation for previous reports of supra-optimality.

Using this improved fitting method, we replicated previous observations *Raposo et al., 2012* showing that animals have improved sensitivity (lower $\sigma$) on multisensory vs. unisensory trials

(*Figure 2e*, red curve is steeper than green/blue curves; *Figure 2f*, top). Interestingly, we observed that animals also had a lower lapse probability ($\lambda + \gamma$) on multisensory trials (*Figure 2e*, asymptotes for red curve are closer to 0 and 1; n = 17 rats, 347,537 trials). This was consistently observed across animals (*Figure 2f*, bottom; the probability of lapses on multisensory trials was 0.06 on average, compared to 0.17 on visual, p=1.4e-4 and 0.21 on auditory, p=1.5e-5). We also noticed that compared to unisensory trials, multisensory trials were slightly biased toward high rates. This bias may reflect that animals' decisions do not exclusively depend on the rate of events, but are additionally weakly influenced by the total event count, as has been previously reported on a visual variant of the task (*Odoemene et al., 2018*).

## Uncertainty-guided exploration offers a novel explanation for lapses where traditional explanations fail

What could account for the reduction in lapse probability on multisensory trials? While adding extra parameters to the ideal observer model fit the behavioral data well and accurately captured the reduction in inverse-slope on multisensory trials, this success does not provide an explanation for why lapses are present in the first place, nor why they differ between stimulus conditions.

To investigate this, we examined possible sources of noise that have traditionally been invoked to explain lapses (*Figure 1d*). The first of these explanations is that lapses might be due to a fixed amount of noise added once the decision has been made. These sources of noise could include decision noise due to imprecision (*Findling et al., 2018*) or motor errors (*Wichmann and Hill, 2001*). However, these sources should hinder decisions equally across stimulus conditions (*Figure 3—figure supplement 1b*), which cannot explain our observation of condition-dependent lapse rates (*Figure 2f*).

A second explanation is that lapses arise due to inattention on a small fraction of trials. Inattention would drive the animal to guess randomly, producing lapse rates whose sum should reflect the probability of not attending (*Figure 3a*, Materials and methods). According to this explanation, the lower lapse rate on multisensory trials could reflect increased attention on those trials, perhaps due to their increased bottom-up salience (i.e. two streams of stimuli instead of one). To examine this possibility, we leveraged a multisensory condition that has been used to manipulate perceptual uncertainty without changing salience in rats and humans (*Raposo et al., 2012*). Specifically, we interleaved standard matched-rate multisensory trials with 'neutral' multisensory trials for which the rate of the auditory stimuli ranged from 9 to 16 Hz, while the visual stimuli was always 12 Hz. This rate was so close to the category boundary (12.5 Hz) that it did not provide compelling evidence for one choice or the other (*Figure 3d*, left), thus reducing the information in the multisensory stimulus and increasing perceptual uncertainty on 'neutral' trials. However, since both 'neutral' and 'matched' conditions are multisensory, they should be equally salient, and since they are interleaved, the animal would be unable to identify the condition without actually attending to the stimulus. According to the inattention model, matched and neutral trials should have the same rate of lapses, only differing in their inverse-slope σ (*Figure 3a*, *Figure 3—figure supplement 1c*).

Contrary to this prediction, we observed higher lapse rates in the 'neutral' condition, where trials had higher perceptual uncertainty on average, compared to the 'matched' condition (*Figure 3d*). This correlation between the average perceptual uncertainty in a condition and its frequency of lapses was reminiscent of the correlation observed while comparing unisensory and multisensory trials (*Figure 2e,f*; *Figure 3—figure supplement 1e*).

Having observed that traditional explanations of lapses fail to account for the behavioral observations, we re-examined a key assumption of ideal observer models used in perceptual decision-making – that subjects have complete knowledge about the rules and rewards (*Dayan and Daw, 2008*). In general, this assumption may not hold true for a number of reasons – even when the stimulus category is known with certainty, subjects might have uncertainty about the values of different actions because they are still in the process of learning (*Law and Gold, 2009*), because they incorrectly assume that their environment is nonstationary (*Yu and Cohen, 2008*), or because they forget over time (*Gershman, 2015*; *Drugowitsch and Pouget, 2018*). In such situations, rather than always 'exploiting' (i.e. picking the action currently assumed to have the highest value), it is advantageous to 'explore' (i.e. occasionally pick actions whose value the subject is uncertain about), in order to gather more information and maximize reward in the long term (*Dayan and Daw, 2008*).

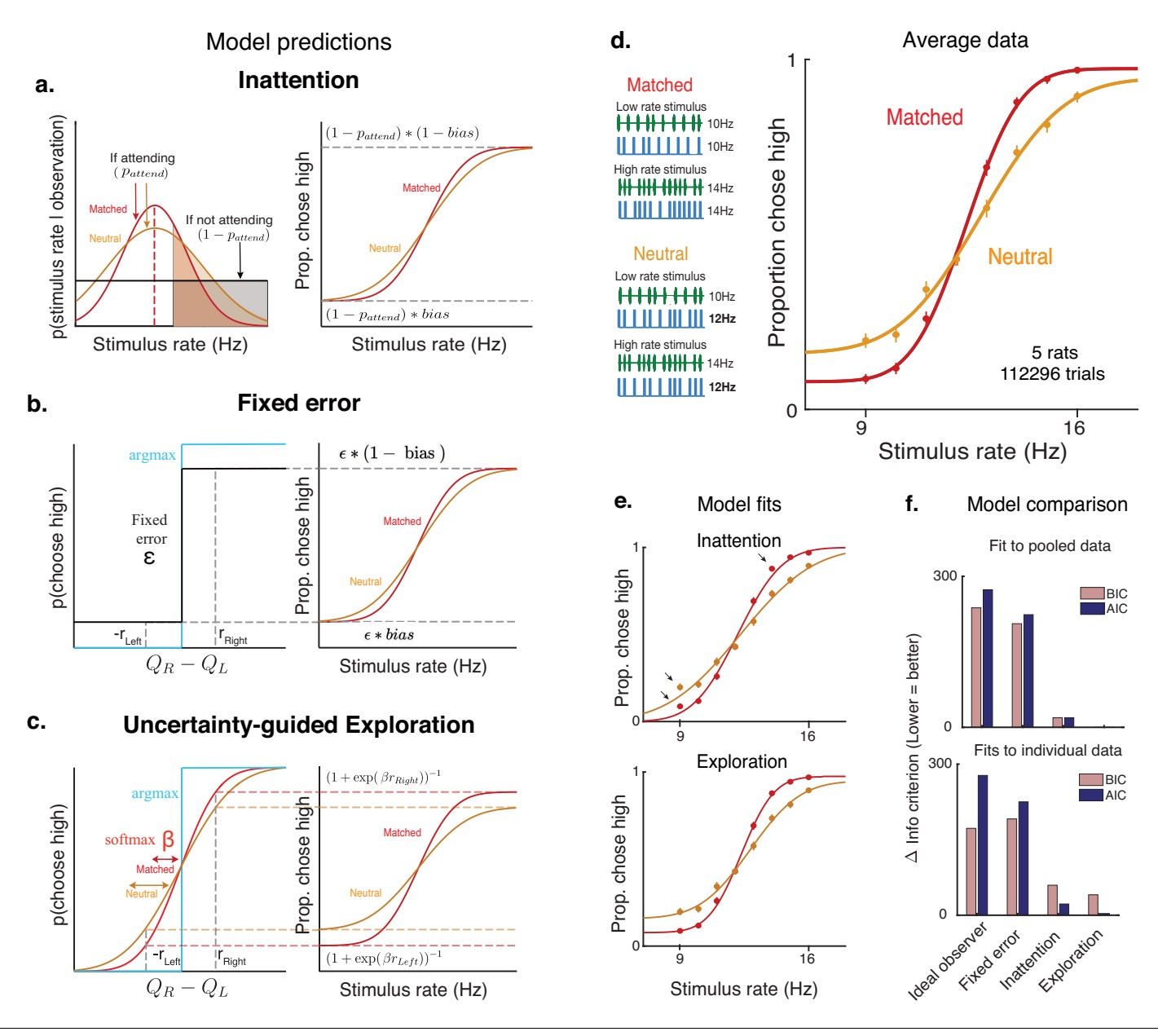

**Figure 3.** Uncertainty-guided exploration offers a novel explanation for lapses. Models of lapses in decision-making: (a) Inattention model of lapses. Left panel: Observer's posterior belief about rate. On a large fraction of trials given by $p_{attend}$, the observer attends to the stimulus and has a peaked belief about the rate whose width reflects perceptual uncertainty (red curve on matched trials, orange curve on neutral trials), but on a small fraction of trials given by $1 - p_{attend}$, the observer does not attend to the stimulus (black curve), leading to equal posterior beliefs of rates being high or low (shaded, clear regions of curves respectively) and guesses according to the probability *bias*, giving rise to lapses (right panel). The sum of lapse rates then reflects $1 - p_{attend}$, while their ratio reflects the *bias*. Since matched and neutral trials are equally salient, they are expected to have the same $p_{attend}$ and hence similar overall lapse rates. (b) Fixed error model of lapses. Lapses could arise due to motor errors occurring on ε fraction of trials, or from decision rules that explore on a fixed proportion ε of trials (black), rather than always maximizing reward (blue). The sum of lapses reflects ε while their ratio reflects any *bias* in motor errors or exploration, leading to a fixed rate of lapses across conditions. (c) Uncertainty-guided exploration model. Lapses can also arise from more sophisticated exploratory decision rules such as the 'softmax' decision rule. Since the difference in expected value from right and left actions ($Q_R - Q_L$) is bounded by the maximum reward magnitudes $r_{Right}$ and $r_{Left}$, even when the stimulus is very easy, the maximum probability of choosing the higher value option is not 1, giving rise to lapses. Lapse rates on either side are then proportional to the reward magnitude on that side, and to a 'temperature' parameter β that is modulated by the uncertainty in action values. Conditions with higher overall perceptual uncertainty (e.g. neutral, orange) are expected to have higher value uncertainty, and hence higher lapses. (d) Left: multisensory stimuli designed to distinguish between attentional and non-attentional sources of lapses. Standard multisensory stimuli with matched visual and auditory rates (top) and 'neutral' stimuli where one modality has a rate very close to the category boundary and is uninformative (bottom). Both stimuli are multisensory and

*Figure 3 continued on next page*

*Figure 3 continued*

designed to have equal bottom-up salience, and can only be distinguished by attending to them and accumulating evidence. Right: rat performance on interleaved matched (red) and neutral (orange) trials. (e) Model fits (solid lines) overlaid on average data points. Deviations from model fits are denoted with arrows. The exploration model (bottom) provides a better fit than the inattention model (top), since it predicts higher lapse rates on neutral trials (orange). (f) Model comparison using Bayes Information Criterion (pink) and Akaike Information Criterion (blue) both favor the uncertainty-guided exploration model for pooled data (top) as well as individual subject data (bottom).

The online version of this article includes the following figure supplement(s) for figure 3:

**Figure supplement 1.** Uncertainty-dependent exploration is the only model that accounts for behavioral data from all three manipulations.
**Figure supplement 2.** Thompson sampling, which balances exploration and exploitation, predicts lapses that increase with perceptual noise.
**Figure supplement 3.** Uncertainty guided exploration outperforms competing models for average and individual data.

Exploratory choices of the lower value action for the easiest stimuli would resemble lapses, and the sum of lapses would reflect the overall degree of exploration.

Choosing how often to explore is challenging and requires trading off immediate rewards for potential gains in information – random exploration would reward subjects at chance, but would reduce uncertainty uniformly about the value of all possible stimulus-action pairs, while a greedy policy (i.e. always exploiting) would yield many immediate rewards while leaving lower value stimulus-action pairs highly uncertain (*Figure 3—figure supplement 2a,b*). Policies that explore randomly on a small fraction of trials (e.g. 'ε-Greedy' policies) do not make prescriptions about how often the subject should explore, and are behaviorally indistinguishable from motor errors when the fraction is fixed (*Figure 3b*). One elegant way to automatically balance exploration and exploitation is to explore more often when one is more uncertain about action values. In particular, a form of uncertainty-guided exploration called Thompson sampling is asymptotically optimal in many general environments (*Leike et al., 2016*), achieving lower regret than other forms of exploration (*Figure 3—figure supplement 2c*). This can be thought of as a dynamic 'softmax' policy (*Figure 3c*), whose 'inverse temperature' parameter ($\beta$) scales with uncertainty (*Gershman, 2018*). This predicts a lower $\beta$ when values are more uncertain, encouraging more exploration and more frequent lapses, and a higher $\beta$ when values are more certain, encouraging exploitation. The limiting case of perfect knowledge ($\beta \rightarrow \infty$) reduces to the reward-maximizing ideal observer.

Subjects' uncertainty about stimulus-action values is compounded by perceptual uncertainty – on trials where the stimulus category is not fully known, credit cannot be unambiguously assigned to one stimulus-action pair when rewards are obtained and value uncertainty is only marginally reduced. Hence conditions where trials have higher perceptual uncertainty on average (e.g. unisensory or neutral trials) will have more overlapping value beliefs, encouraging more exploration and giving rise to more frequent lapses (*Figure 3—figure supplement 2d*).

As a result, on neutral multisensory trials, the uncertainty-guided exploration model predicts an increase not only in the inverse slope parameter $\sigma$, but also in the rate of lapses, just as we observed (*Figure 3d*). In fact, this model predicts that both slope and lapse parameters on neutral trials should match those on auditory trials, since these conditions have comparable levels of perceptual uncertainty. The data was well fit by the exploration model (*Figure 3e*, bottom) and satisfied both predictions (*Figure 4—source data 1*, Neutral has higher $\sigma$ and lower $\beta$ than Multisensory, and matched $\sigma$ and $\beta$ to Auditory) . By contrast, the inattention model predicts that both conditions would have the same lapse rates, with the neutral condition simply having a larger inverse slope $\sigma$. This model provided a worse fit to the data, particularly missing the data at extreme stimulus values where lapses are most clearly apparent (*Figure 3e*, top). Model comparison using BIC and AIC favored the exploration model over the inattention model, both for fits to pooled data across subjects (*Figure 3f*, top) and fits to individual subject data (*Figure 3f*, bottom, *Figure 3—figure supplement 3*, for the 3/5 subjects rejected by ideal observer model that is, with sizable lapses. Both predictions of the exploration model were confirmed using unconstrained descriptive fits to individuals, and held up for 4/5 subjects).

To further understand the precise relationship between perceptual uncertainty and lapses under this form of exploration, we simulated learning in a Thompson sampling agent for various levels of sensory noise, and found a roughly linear relationship between sensory noise and average lapse rate. Hence we fit a constrained version of the exploration model to the multisensory data from 17 rats, where the degree of exploratory lapses was constrained to be a linear function of that

condition's sensory noise (with two free parameters – slope and intercept, rather than three free parameters for the three conditions). This model yielded lower BIC than the unconstrained exploration model in all 14/17 rats that were rejected by the ideal observer model (*Figure 3—figure supplement 3c*), and yielded similar slope and intercept parameters across animals (*Figure 3—figure supplement 2e*).

## Reward manipulations confirm predictions of exploration model

One of the key claims of the uncertainty-guided exploration model is that lapses are exploratory choices made with full knowledge of the stimulus, and should therefore depend only on the expected rewards associated with that stimulus category (*Figure 3—figure supplement 2*). This is in stark contrast to the inattention model and many other kinds of disengagement (*Figure 4—figure supplement 1*), according to which lapses are caused by the observer disregarding the stimulus, and hence lapses at the two extreme stimulus levels are influenced by a common underlying guessing process that depends on expected rewards from both stimulus categories. This is also in contrast to fixed error models such as motor error or $\epsilon$-greedy models in which lapses are independent of expected reward (*Figure 3b*).

Therefore, a unique prediction of the exploration model is that selectively manipulating expected rewards associated with one of the stimulus categories should only change the explore–exploit tradeoff for that stimulus category, selectively affecting lapses at one extreme of the psychometric function. Conversely, inattention and other kinds of disengagement predict that both lapses should be affected, while fixed error models predict that neither should be affected (*Figure 4a*, *Figure 3—figure supplement 1*, *Figure 4—figure supplement 1*).

To experimentally test these predictions, we tested rats on the rate discrimination task with asymmetric rewards (*Figure 4b*, top). Instead of rewarding high- and low-rate choices equally, we increased the water amount on the reward port associated with high rates (rightward choices) so it was 1.5 times larger than before, without changing the reward on the low-rate side (leftward choices). In a second rat cohort we did the opposite: we devalued the choices associated with high-rate trials by decreasing the water amount on that side port, and so it was 1.5 times smaller than before, without changing the reward on the low-rate side.

The animals' behavior on the asymmetric-reward task matched the predictions of the exploration model. Increasing the reward size on choices associated with high rates led to a decrease in lapses for the highest rates and no changes in lapses for the lowest rates (*Figure 4c*, left; n = 3 rats, 6976 trials). Decreasing the reward of choices associated with high rates led to an increase in lapses for the highest rates and no changes in lapses for the lower rates (*Figure 4c*, right; n = 3 rats, 11,164 trials). This shows that both increasing and decreasing the value of actions associated with one of the stimulus categories selectively affects lapses on that stimulus category, unlike the predictions of the inattention model.

A key claim of the uncertainty-guided exploration model is that the effects of reward manipulations on lapses arise from a selective shift in the trade-off between exploiting the most rewarding action and exploring uncertain ones, rather than from a non-selective bias toward the side with bigger rewards. Importantly, the model predicts that in the absence of uncertainty, decisions should be perfectly exploitative and unaffected by reward imbalances, since subjects would always be comparing perfectly certain, nonzero rewards to zero. To determine whether the effects that we observed were truly driven by uncertainty, we examined performance on randomly interleaved 'sure bet' trials on which the uncertainty was very low (*Figure 4b*, bottom). On these trials, a pure tone was played during the fixation period, after which an LED at one of the side ports was clearly illuminated, indicating a reward. Sure-bet trials comprised 6% of the total trials, and as with the rate discrimination trials, left and right trials were interleaved. Owing to the low perceptual uncertainty and consequently low value uncertainty, the model predicts that animals would quickly reach an 'exploit' regime, achieving perfect performance on these trials. Importantly, our model predicts that performance on these 'sure-bet' trials would be unaffected by imbalances in reward magnitude, since the 'exploit' action remains unchanged.

In keeping with this prediction, performance on sure-bet trials was near perfect (rightward probabilities of 0.003 [0.001,0.01] and 0.989 [0.978,0.995] on go-left and go-right trials respectively) and unaffected following reward manipulations (*Figure 4d*: Rightward probabilities of 0.004 [0.001, 0.014] and 0.996 [0.986,0.999] on increased reward, 0.006 [0.003,0.012] and 0.99 [0.983,0.994] on

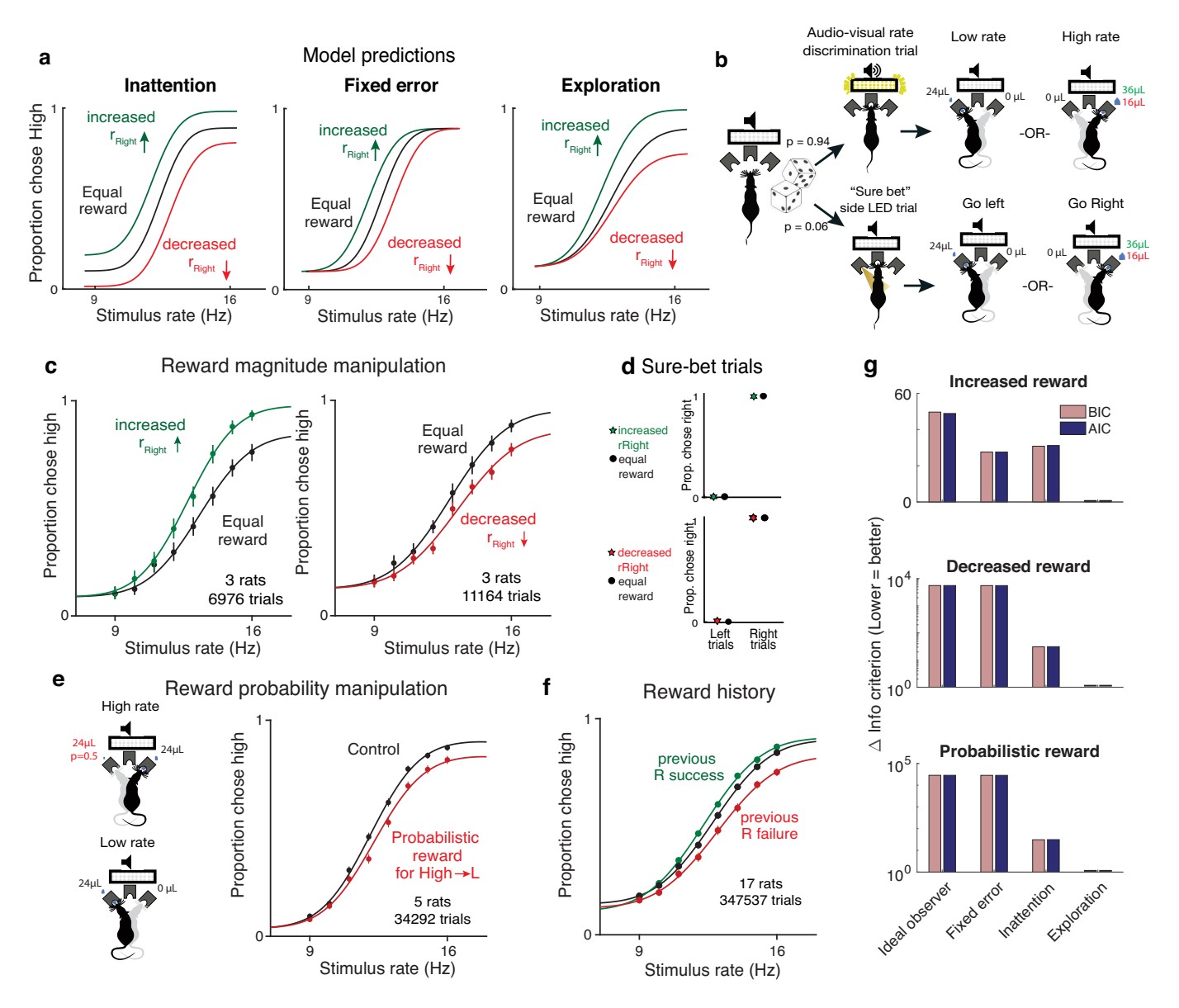

**Figure 4.** Reward manipulations match predictions of the exploration model. (a) The inattention, fixed error, and exploration models make different predictions for increases and decreases in the reward magnitude for rightward (high-rate) actions. The inattention model (left panel) predicts changes in lapses for both high- and low-rate choices, while fixed error models such as motor error or ε-greedy (center) predict changes in neither lapse, and the uncertainty-dependent exploration model (right) predicts changes in lapses only for high-rate choices. Black line denotes equal rewards on both sides; green, increased rightward reward; red, decreased rightward reward. (b) Schematic of rate discrimination trials and interleaved 'sure bet' trials. The majority of the trials (94%) were rate discrimination trials as described in *Figure 1*. On sure-bet trials, a pure tone was played during a 0.2 s fixation period and one of the side ports was illuminated once the tone ended to indicate that reward was available there. Rate discrimination and sure-bet trials were randomly interleaved, as were left and right trials, and the rightward reward magnitude was either increased (36 µL) or decreased (16 µL) while maintaining the leftward reward at 24 µL. (c) Rats' behavior on rate discrimination trials following reward magnitude manipulations. High-rate lapses decrease when water reward for high-rate choices is increased (left panel; n = 3 rats, 6976 trials), while high-rate lapses increase when reward on that side is decreased (right panel; n = 3 rats, 11,164 trials). Solid curves are exploration model fits with a single parameter change accounting for the manipulation. (d) Rats show nearly perfect performance on sure-bet trials and are unaffected by reward manipulations on these trials. (e) Reward probability manipulation. (Left) Schematic of probabilistic reward trials, incorrect (leftward) choices on high rates were rewarded with a probability of 0.5, and all other rewards were left unchanged. (Right) Rats' behavior and exploration model fits showing a selective increase in high-rate lapses (n = 5 rats, 34,292 trials). (f) Rats' behavior on equal reward trials conditioned on successes (green) or failures (red) on the right on the previous trials resembles effects of reward size manipulations. (g) Model comparison showing that Akaike Information Criterion and Bayes Information Criterion both favor the exploration model on data from all three manipulations.

*Figure 4 continued on next page*

*Figure 4 continued*

The online version of this article includes the following source data and figure supplement(s) for figure 4:

**Source data 1.** Fit parameters to pooled data across rats.
**Figure supplement 1.** Alternative models of inattentional lapses.
**Figure supplement 2.** Psychometric functions with lapses make it possible to assign perturbations effects to specific stages of decision-making.

decreased reward). This suggests that the effects of reward manipulations that we observed (*Figure 4c*) are not a default consequence of reward imbalance, but a consequence of a reward-dependent trade-off between exploitation and uncertainty-guided exploration.

As an additional test of the model, we manipulated expected rewards by probabilistically rewarding incorrect choices for one of the stimulus categories. Here, leftward choices on high-rate ('go right') trials were rewarded with a probability of 0.5, while leaving all other rewards unchanged (*Figure 4e* left). The exploration model predicts that this should selectively increase the value of leftward actions on high-rate trials, hence shifting the trade-off toward exploration on high rates and increasing high-rate lapses. Indeed, this is what we observed (*Figure 4e* right, n = 5 animals, 347,537 trials), and the effect was strikingly similar to the decreased reward experiment, even though the two manipulations affect high-rate action values through changes on opposite actions. This experiment in particular distinguishes the exploration model from motivation-dependent models of disengagement or inattention in which overall reward modulates the total lapse rate through a nonspecific process that averages over stimulus categories (*Figure 4—figure supplement 1a–c,f*). Moreover, this suggests that lapses reflect changes in stimulus-specific action value caused by changing either reward magnitudes or reward probabilities, as one would expect from the exploration model. Across experiments (*Figure 4—source data 1*) and individuals, these changes were captured by selectively changing the relevant baseline action value in the model, despite variability in these baselines.

An added consequence of uncertainty in action values is that it should encourage continued learning even in the absence of explicit reward manipulations. This means that animals should continue to use the outcomes of previous trials to update the values of different actions as long as this uncertainty persists. Such persistent learning has been observed in a number of studies (*Busse et al., 2011*; *Lak et al., 2018*; *Mendonca et al., 2018*; *Odoemene et al., 2018*; *Pinto et al., 2018*; *Scott et al., 2015*). The uncertainty-dependent exploration model predicts that the effect of recent outcome history on action values should manifest as changes in lapse rates, rather than as horizontal biases caused by irrelevant, non-sensory evidence as is often assumed (*Busse et al., 2011*). For example, the action value of rightward choices should increase following a rightward success, producing similar changes to lapses as increased rightward reward magnitude. As predicted, trials following rewarded and unrewarded rightward choices showed decreased and increased lapses, respectively (*Figure 4f*; same rats and trials as in *Figure 2e*). Taken together, manipulations of value confirm the predictions of the uncertainty-dependent exploration model (*Figure 4g*).

## Lapses are a powerful tool for assigning decision-related computations to neural structures based on disruption experiments

The results of the behavioral manipulations (above) predict that unilateral disruption of neural regions that leads to a one-sided scaling of learnt stimulus-action values should affect lapse rates asymmetrically. In contrast, disruptions to areas that process sensory evidence would lead to horizontal biases without affecting action values or lapses, and disruptions to motor areas that make one of the actions harder to perform irrespective of the stimulus would affect both lapses (*Figure 4—figure supplement 2a* top, middle). Crucially, in the absence of lapses, all three of these disruptions would drive an identical behavioral effect, a horizontal shift of the psychometric function (*Figure 4—figure supplement 2a* bottom). Indeed, the same reward manipulations that gave rise to distinct value biases in rats with sizeable lapses (*Figure 4—figure supplement 2b* top) led to horizontal shifts indistinguishable from sensory biases in highly trained rats with negligible lapses on multisensory trials (*Figure 4—figure supplement 2b* bottom). This suggests that lapses are actually informative about decision-making computations and can be used as a tool to determine which computations are affected by disruptions of a candidate brain region. To demonstrate this, we

identified two candidate areas, secondary motor cortex (M2) and posterior striatum (pStr), that receive convergent input from primary visual and auditory cortices (*Figure 5—figure supplement 1*, results of simultaneous anterograde tracing from V1 and A1; also see *Jiang and Kim, 2018*; *Barthas and Kwan, 2017*). In previous work, disruptions of these areas had effects on auditory decisions, including changes in lapses (*Erlich et al., 2015*; *Guo et al., 2018*). However, considerable controversy remains as to which computations were affected by those disruptions. The effects were largely interpreted in terms of traditional ideal observer models (see *Siniscalchi et al., 2019* for a notable exception), and thus attributed to perceptual biases (*Guo et al., 2018*), leaky accumulation (*Erlich et al., 2015*) or post-categorization biases (*Piet et al., 2017*; *Erlich et al., 2015*). Notably, the asymmetric effects on lapses seen in these studies resembled the effects of the reward manipulations in our task, hinting that they may actually arise from action value changes. Importantly, these existing studies used only auditory stimuli, so were limited in their ability to distinguish sensory-specific deficits from action value deficits.

Here, we used analyses of lapses to determine the decision-related computations altered by unilateral disruption of M2 and pStr. If these disruptions affected action values, the exploration model makes three strong predictions. First, because action values are computed late in the decision-making process, the model predicts that the effects should not depend on the modality of the stimulus. We therefore performed disruptions in animals doing interleaved auditory, visual, and multisensory trials. If pStr and M2 indeed compute action value, then following unilateral disruption of these areas, our model should capture changes to all three modalities by a single parameter change to the contralateral action value. Second, these disruptions should selectively affect lapses on stimuli associated with contralateral actions, irrespective of the stimulus-response contingency. To test this, we performed disruptions on animals trained on standard and reversed contingencies. Finally, because altered action values should have no effect when there is no uncertainty and consequently no exploration, disruption to pStr and M2 should spare performance on sure-bet trials (*Figure 4b*, bottom).

We suppressed activity of neurons in each of these areas using muscimol, a GABAA agonist, during our multisensory rate discrimination task. We implanted bilateral cannulae in M2 (*Figure 5a*, *Figure 5—figure supplement 2b*; n = 5 rats; +2 mm AP 1.3 mm ML, 0.3 mm DV) and pStr (*Figure 5a*, *Figure 5—figure supplement 2a*; n = 6 rats; −3.2 mm AP, 5.4 mm ML, 4.1 mm DV). On control days, rats were infused unilaterally with saline, followed by unilateral muscimol infusion the next day (M2: 0.1–0.5 µg, pStr 0.075–0.125 µg). We compared performance on the multisensory rate discrimination task for muscimol days with preceding saline days. Inactivation of the side associated with low-rate choices biased the animals to make more low-rate choices (*Figure 5b*; left six panels: empty circles, inactivation sessions; full circles, control sessions), while inactivation of the side associated with high rates biased them to make more high-rate choices (*Figure 5b*, right six panels). The inactivations largely affected lapses on the stimulus rates associated with contralateral actions, while sparing those associated with ipsilateral actions (*Figure 5c*). These results recapitulated previous findings and were strikingly similar to the effects we observed following reward manipulations (as seen in *Figure 4c*, right panel). These effects were seen across areas (*Figure 5b*, top, M2; bottom, pStr) and modalities (*Figure 5b*; green, auditory; blue, visual and red, multisensory).

Fitting averaged data across rats with the exploration model revealed that, in keeping with the first model prediction, the effects on lapses in all modalities could be captured by scaling the contralateral action value by a single parameter (*Figure 5b*, joint fits to control [solid lines] and inactivation trials [dotted lines] across modalities with the 'biased value' model, differing only by a single parameter), similar to the reward manipulation experiments. Animals that were inactivated on the side associated with high rates showed increased lapses on low-rate trials (*Figure 5c*, bottom right; data points are above the unity line; n = 9 rats), but unchanged lapses on high-rate trials (*Figure 5c*, top right; data points are on the unity line). This was consistent across areas and modalities (*Figure 5c*; M2, triangles; pStr, circles; blue, visual; green, auditory; red, multisensory). Similarly, animals that were inactivated on the side associated with low rates showed the opposite effect: increased lapses on high-rate trials (*Figure 5c*, top left; n = 10 rats), while lapses did not change for low-rate trials (*Figure 5c*, bottom left). Fits to individual animals revealed that the majority of animals were best fit by the 'biased value' model (6/8 rats in M2 – *Figure 5—figure supplement 3*, 7/11 in pStr – *Figure 5—figure supplement 4*), and the remaining animals were best fit by the 'biased effort' model.

In keeping with the second prediction, when we compared the effects of the disruptions in animals trained on standard and reversed contingencies (low rates rewarded with leftward or rightward

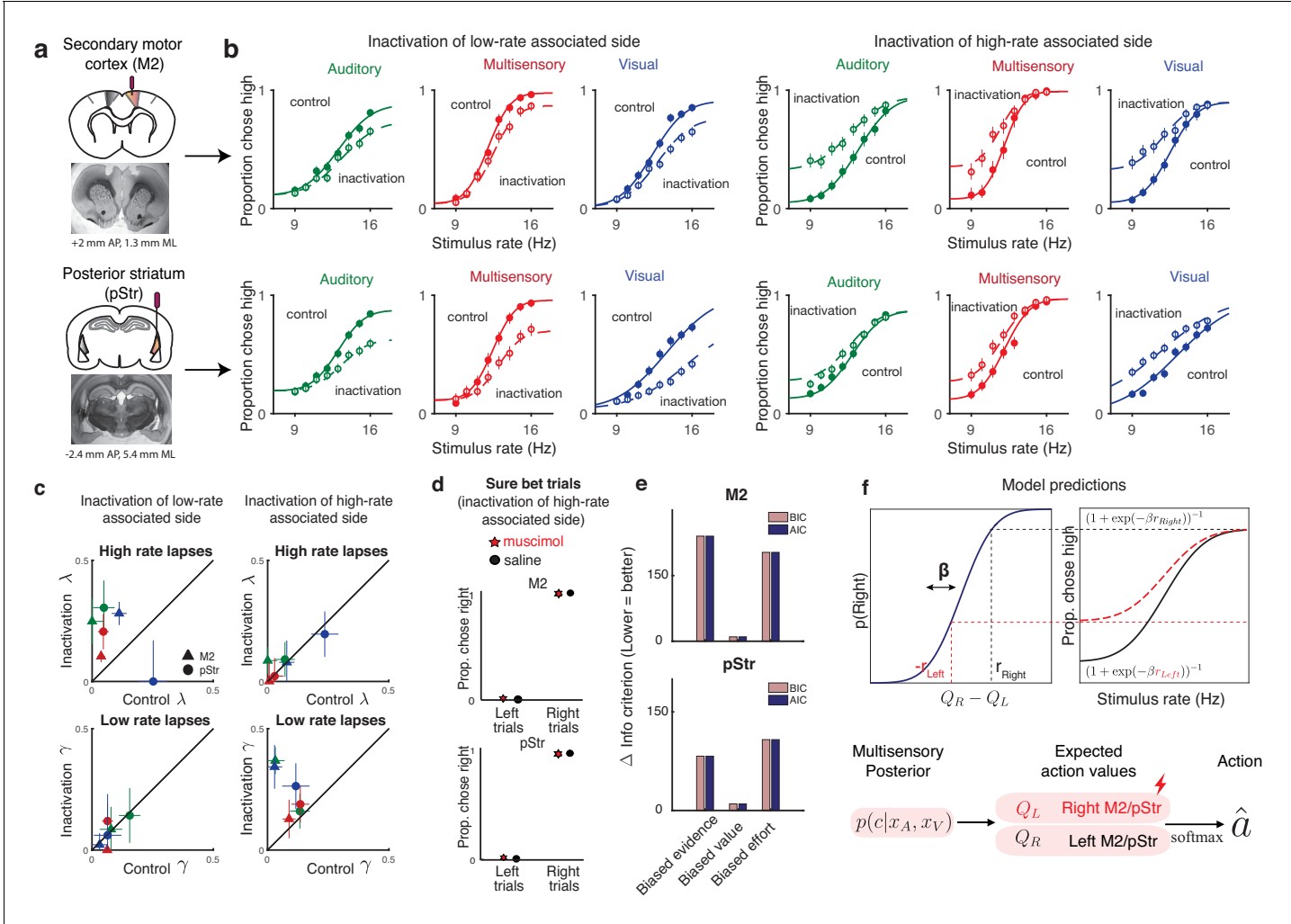

**Figure 5.** Inactivation of secondary motor cortex and posterior striatum affects lapses, suggesting a role in action value encoding. (**a**) Schematic of cannulae implants in M2 (top) and pStr (bottom) and representative coronal slices. For illustration purposes only, the schematic shows implants in the right hemisphere; however, the inactivations shown in panel (**b**) were performed unilaterally on both hemispheres. (**b**) Unilateral inactivation of M2 (top) and pStr (bottom). Left six plots: inactivation of the side associated with low rates shows increased lapses for high rates on visual (blue), auditory (green), and multisensory (red) trials (M2: n = 5 rats; 10,329 control trials, full line; 6174 inactivation trials, dotted line; pStr: n = 5 rats; 10,419 control trials; 6079 inactivation trials). Right six plots: inactivation of the side associated with high rates shows increased lapses for low rates on visual, auditory, and multisensory trials (M2: n = 3 rats; 5678 control trials; 3816 inactivation trials; pStr: n = 6 rats; 11,333 control trials; 6838 inactivation trials). Solid lines are exploration model fits, accounting for inactivation effects across all three modalities by scaling all contralateral values by a single parameter. (**c**) Increased high-rate lapses following unilateral inactivation of the side associated with low rates (top left); no change in low-rate lapses (bottom left) and vice versa for inactivation of the side associated with high rates (top, bottom right). Control data on the abscissa is plotted against inactivation data on the ordinate. Same animals as in b. Green, auditory trials; blue, visual trials; red, multisensory trials. Abbreviations: posterior striatum (pStr), secondary motor cortex (M2). (**d**) Sure bet trials are unaffected following inactivation. Pooled data shows that rats that were inactivated on the side associated with high rates make near perfect rightward and leftward choices Top, M2 (three rats); bottom, pStr (six rats). (**e**) Model comparison of three possible multisensory deficits – reduction of contralateral evidence by a fixed amount (left), reduction of contralateral value by a fixed amount (center), or an increased contralateral effort by a fixed amount (right). Both Akaike Information Criterion and Bayes Information Criterion suggest a value deficit. (**f**) Proposed computational role of M2 and striatum. Lateralized encoding of left and right action values by right and left M2/pStr (bottom) explains the asymmetric effect of unilateral inactivations on lapses (top).

The online version of this article includes the following figure supplement(s) for figure 5:

**Figure supplement 1.** pStr and M2 receive direct projections from visual and auditory cortex.

**Figure supplement 2.** Histological slices of implanted rats.

**Figure supplement 3.** Single rat performance following M2 inactivation.

**Figure supplement 4.** Single rat performance following pStr inactivation.

**Figure supplement 5.** Unilateral inactivation of M2 or pStr biases performance ipsilaterally and increases contralateral lapses.

*Figure 5 continued on next page*

*Figure 5 continued*

**Figure supplement 6.** Inactivations devalue contralateral actions irrespective of associated stimulus.

**Figure supplement 7.** No significant effect on movement parameters following muscimol inactivation.

actions respectively), the effects were always restricted to lapses on the stimuli associated with the side contralateral to the inactivation (*Figure 5—figure supplement 5*), always resembling a devaluation of contralateral actions (*Figure 5—figure supplement 6*).

A model comparison across rats revealed that a fixed multiplicative scaling of contralateral value captured the inactivation effects much better than a fixed reduction in contralateral sensory evidence or a fixed addition of contralateral motor effort, both for M2 (*Figure 5e*, top) and pStr (*Figure 5e*, bottom). In uncertain conditions, this reduced contralateral value gives rise to more exploratory choices and hence more lapses on one side (*Figure 5f*, top).

The final prediction of the exploration model is that changes in action value will only affect trials in which there was uncertainty about the outcome. In keeping with that prediction, performance was spared on sure-bet trials (*Figure 5d*): rats made correct rightward and leftward choices regardless of the side that was inactivated. This observation provides further reassurance that the changes we observed on more uncertain conditions did not simply reflect motor impairments that drove a tendency to favor ipsilateral movements. Additional movement parameters such as wait time in the center port and movement times to ipsilateral and contralateral reward ports were likewise largely spared (*Figure 5—figure supplement 7*), suggesting that effects on decision outcome were not due to an inactivation-induced motor impairment.

Together, these results demonstrate that lapses are a powerful tool for interpreting behavioral changes in disruption experiments. For M2 and pStr disruptions, our analysis of lapses and deployment of the exploration model allowed us to reconcile previous inactivation studies. Our results suggest that M2 and pStr have a lateralized, modality-independent role in computing the expected value of actions (*Figure 5f*, bottom).

## Discussion

Perceptual decision-makers have long been known to display a small fraction of errors even on easy trials. Until now, these 'lapses' were largely regarded as a nuisance and lacked a comprehensive, normative explanation. Here, we propose a novel explanation for lapses: that they reflect a strategic balance between exploiting known rewarding options and exploring uncertain ones. Our model makes strong predictions for lapses under diverse decision-making contexts, which we have tested here. First, the model predicts more lapses on conditions with higher perceptual uncertainty, such as unisensory (*Figure 2*) or neutral (*Figure 3*), compared to matched multisensory or sure-bet conditions. Second, the model predicts that stimulus-specific reward manipulations should produce stimulus-specific effects on lapses, sparing decisions about un-manipulated or highly certain stimulus-action pairs (*Figure 4*). Finally, the model predicts that lapses should be affected by perturbations to brain regions that encode action value. Accordingly, we observed that inactivations of secondary motor cortex and posterior striatum affected lapses similarly across auditory, visual and multisensory decisions, and could be accounted for by a one-parameter change to the action value (*Figure 5*). Taken together, our model and experimental data argue strongly that far from being a nuisance, lapses are informative about animals' subjective action values and reflect a trade-off between exploration and exploitation.

Considerations of value have provided many useful insights into aspects of behavior that seem sub-optimal at first glance from the perspective of perceptual ideal observers. For instance, many perceptual tasks are designed with accuracy in mind – defining an ideal observer as one who maximizes accuracy, in line with classical signal detection theory. However, in practice, the success or failure of different actions may be of unequal value to subjects, especially if reward or punishment is delivered explicitly, as is often the case with nonhuman subjects. This may give rise to biases that can only be explained by an observer that maximizes expected utility (*Dayan and Daw, 2008*). Similarly, outcomes on a given trial can influence decisions about stimuli on subsequent trials through reinforcement learning, giving rise to serial biases. These biases occur even though the ideal

observer should treat the evidence on successive trials as independent (*Lak et al., 2018*; *Mendonca et al., 2018*). When subjects can control how long they sample the stimulus, subjects maximizing reward rate may choose to make premature decisions, sacrificing accuracy for speed (*Bogacz et al., 2006*; *Drugowitsch et al., 2014*). Finally, additional costs of exercising mental effort could lead to bounded optimality through 'satisficing' or finding good enough solutions (*Mastrogiorgio and Petracca, 2018*; *Fan et al., 2018*).

Here, we take further inspiration from considerations of value to provide a novel, normative explanation for lapses in perceptual decisions. Our results argue that lapses are not simply accidental errors made as a consequence of attentional 'blinks' or motor 'slips', but can reflect a deliberate, internal source of behavioral variability that facilitates learning and information gathering when the values of different actions are uncertain. This explanation connects a well-known strategy in value-based decision-making to a previously mysterious phenomenon in perceptual decision-making.

Although exploration no longer yields the maximum utility on any given trial, it is critical for environments in which there is uncertainty about expected reward or stimulus-response contingency, especially if these need to be learnt or refined through experience. By encouraging subjects to sample multiple options, exploration can potentially improve subjects' knowledge of the rules of the task, helping them to increase long-term utility. This offers an explanation for the higher rate of lapses seen in humans on tasks with abstract (*Raposo et al., 2012*), non-intuitive (*Mihali et al., 2018*), or non-verbalizable (*Flesch et al., 2018*) rules. Exploration is also critical for dynamic environments in which rules or rewards drift or change over time. Subjects adapted to such dynamic real-world environments might entertain the possibility of non-stationarity even in tasks or periods where rewards are truly stationary, and such mismatched beliefs predict residual levels of exploration even in well-trained subjects (*Figure 3—figure supplement 2g*, middle). Such beliefs could be probed by challenging subjects with unsignalled changes in rewards and measuring how quickly they recover from these change-points. For instance, primates with higher levels of tonic exploration on cognitive set-shifting tasks (*Ebitz et al., 2019*) are more flexible and make fewer perseverative errors at change-points, at the cost of more lapses in rule adherence during stable periods.

Balancing exploration and exploitation is computationally challenging, and the mechanism we propose here, Thompson sampling, is an elegant heuristic for achieving this balance. This strategy has been shown to be utilized by humans in value-based decision-making tasks (*Wilson et al., 2014*; *Speekenbrink and Konstantinidis, 2015*; *Gershman, 2018*) and is asymptotically optimal even in partially observable environments involving perceptual uncertainty such as ours (*Figure 3—figure supplement 2c*, *Leike et al., 2016*). It can be naturally implemented through a sampling scheme where the subject samples action values from a learnt distribution and then maximizes with respect to the sample. This strategy predicts that conditions with higher perceptual uncertainty and consequently higher value uncertainty should have more exploration, and consequently higher lapse rates, explaining the pattern of lapse rates we observed on unisensory vs. multisensory trials as well as on neutral vs. matched trials. A lower rate of lapses on multisensory trials has also been reported on a visual-tactile task in rats (*Nikbakht et al., 2018*) and a vestibular integration task in humans (*Bertolini et al., 2015*) and can potentially account for the apparent supra-optimal integration that has been reported in a number of rodent, nonhuman primate and human studies (*Nikbakht et al., 2018*; *Hou et al., 2018*; *Raposo et al., 2012*). A strong prediction of uncertainty guided exploration is that the animal should quickly learn to exploit on conditions with little or no uncertainty, as we observed on sure-bet trials (*Figures 4d* and *5d*).

Uncertainty-guided exploration also predicts that exploratory choices, and consequently lapses, should decrease with training as the animal becomes more certain of the rules and expected rewards, explaining training-dependent effects on lapses in our rats (*Figure 3—figure supplement 2g,* right) and similar effects reported in primates (*Law and Gold, 2009*; *Cloherty et al., 2019*). This can also potentially explain why children have higher lapse rates (*Witton et al., 2017*; *Manning et al., 2018*), as they have been shown to be more exploratory in their decisions than adults (*Lucas et al., 2014*).

A unique prediction of the exploration model is that one-sided reward manipulations should have one-sided effects on lapses, unlike the inattention or motor error models. These predictions are borne out in our data (*Figure 4c*); moreover, they offer a principled, theoretically grounded way to distinguish between different sources of lapses. This approach can be extended to connect richer statistical descriptions of behavior to psychological variables such as evidence and action value. For

instance, some authors have proposed that some of the variance attributed to lapses can be accounted for by allowing psychometric parameters to drift across trials (*Roy et al., 2018*) or switch between different settings (*Ashwood et al., 2019*). Whether this parametric non-stationarity arises from non-stationary evidence weighting across trials caused by inattention, variable attention (*Shen and Ma, 2019*) or attention to irrelevant evidence (*Busse et al., 2011*), or whether it arises from non-stationary beliefs about action values that encourage continued learning (*Lak et al., 2018*) and bouts of exploration (*Ebitz et al., 2019*) can be tested using one-sided reward manipulations, and by extending our model to include trial-by-trial updates of action value based on the history of evidence and outcomes (*Pisupati et al., 2019*). By decoupling the values of different actions on the two stimulus categories, one-sided reward manipulations distinguish between incorrect decisions made due to a lack of knowledge about the stimulus category (i.e. inattention) and those made despite this knowledge, due to uncertainty about action values (i.e. exploration). An alternative way to decouple these two kinds of errors would be to offer subjects additional actions, e.g. by adding explicit 'opt-out' actions (*Zatka-Haas et al., 2019*), or by adding task-irrelevant actions that subjects need to learn to avoid (*Mihali et al., 2018*), affording more opportunities to distinguish exploratory and inattentive decisions than tasks with two alternative actions.

In addition to diagnosing or remedying lapses, the exploration model can be used to harness lapses to pinpoint decision-making computations in the brain. Our model suggests that the asymmetric effects on lapses seen during unilateral inactivations of prefrontal and striatal regions (*Figure 5b*) arise from a selective devaluation of learnt contralateral stimulus-action values. This interpretation reconciles a number of studies that have found asymmetric effects of inactivating these areas during perceptual decisions (*Erlich et al., 2015*; *Zatka-Haas et al., 2019*; *Wang et al., 2018*; *Guo et al., 2018*) with their established roles in encoding action value (*Sul et al., 2011*) during value-based decisions , and strengthens previous proposals that these areas arbitrate between perceptual and value-based influences on decisions (*Lee et al., 2015*; *Barthas and Kwan, 2017*; *Siniscalchi et al., 2019*). The effects of inactivation in these studies are consistent with a 'devaluation' deficit, or multiplicative scaling of learnt stimulus-action values, resembling the majority of our inactivations (6/8 rats in M2, 7/11 in pStr) and selectively affecting lapses on stimuli strongly associated with the devalued actions. However, inactivations sometimes resembled additive deficits in action value (2/8 rats in M2, 4/11 in pStr), akin to an added 'effort' in performing the associated action irrespective of its learnt value, consistent with some reports in striatum (*Tai et al., 2012*). Further work will be needed to precisely understand the nature of value representations in these regions and why they are sometimes multiplicatively and sometimes additively impacted by inactivations.

An open question that remains is how the brain might tune the degree of exploration in proportion to uncertainty. An intriguing candidate for this is dopamine, whose phasic responses have been shown to reflect state uncertainty (*Starkweather et al., 2017*; *Babayan et al., 2018*; *Lak et al., 2018*), and whose tonic levels have been shown to modulate exploration in mice on a lever-press task (*Beeler et al., 2010*), and context-dependent song variability in songbirds (*Leblois et al., 2010*). Dopaminergic genes have been shown to predict individual differences in uncertainty-guided exploration in humans (*Frank et al., 2009*), and dopaminergic disorders such as Parkinson's disease have been shown to disrupt the uncertainty-dependence of lapses across conditions on a multisensory task (*Bertolini et al., 2015*), while L-Dopa, a Parkinson's drug and dopamine precursor, has been shown to attenuate uncertainty-guided exploration (*Chakroun et al., 2019*). Patients with ADHD, another disorder associated with dopaminergic dysfunction, have been shown to display both increased perceptual variability and increased task-irrelevant motor output, a measure that correlates with lapses (*Mihali et al., 2018*). Finally, tonic exploration and lapses of rule adherence are reduced in nonhuman primates that are administered cocaine (*Ebitz et al., 2019*), which interferes with dopamine transport. A promising avenue for future studies is to leverage the informativeness of lapses and the precise control of uncertainty afforded by multisensory tasks, in conjunction with perturbations or recordings of dopaminergic circuitry, to further elucidate the connections between perceptual and value-based decision-making systems.

# Materials and methods

**Key resources table**

| Reagent type (species) or resource | Designation | Source or reference | Identifiers | Additional information |
|---|---|---|---|---|
| Strain, strain background (*Rattus norvegicus* domestica, male and female) | Long-Evans Rat | Taconic Farms | SimTac:LE | TAC: LONGEV-M, TAC: LONGEV-F |
| Recombinant DNA reagent | AAV2.CB7.CI. EGFP.WPRE.RBG | UPenn Vector Core | | Obtained from the laboratory of Dr. Partha Mitra at CSHL |
| Recombinant DNA reagent | AAV2.CAG.td Tomato.WPRE.SV40 | UPenn Vector Core | | Obtained from the laboratory of Dr. Partha Mitra at CSHL |
| Chemical compound, drug | Muscimol | abcam | ab120094 | |
| Software, algorithm | PALAMEDES toolbox | *Prins and Kingdom, 2018* | | doi: 10.3389/fpsyg.2018.01250 |
| Software, algorithm | MATLAB | The Mathworks, Inc | | |

## Behavior

### Animal subjects and housing

All animal procedures and experiments were in accordance with the National Institutes of Health's Guide for the Care and Use of Laboratory Animals and were approved by the Cold Spring Harbor Laboratory Animal Care and Use Committee. Experiments were conducted with 34 adult male and female Long Evans rats (250–350 g, Taconic Farms) that were housed with free access to food and restricted access to water starting from the onset of behavioral training. Rats were housed on a reversed light–dark cycle; experiments were run during the dark part of the cycle. Rats were pair-housed during the whole training period.

### Animal training and behavioral task

Rats were trained following previously established methods (*Raposo et al., 2012*; *Sheppard et al., 2013*; *Raposo et al., 2014*; *Licata et al., 2017*). Briefly, rats were trained to wait in the center port for 1000 ms while stimuli were presented, and to associate stimuli with left/right reward ports. Stimuli for each trial consisted of a series of events: auditory clicks from a centrally positioned speaker, full-field visual flashes, or both together. Stimulus events were separated by either long (100 ms) or short (50 ms) intervals. For the easiest trials, all inter-event intervals were identical, generating rates that were nine events per second (all long intervals) or 16 events per second (all short intervals). More difficult trials included a mixture of long and short intervals, generating stimulus rates that were intermediate between the two extremes and therefore more difficult for the animal to judge. The stimulus began after a variable delay following when the rats snout broke the infrared beam in the center port. The length of this delay was selected from a truncated exponential distribution ($\lambda$ = 30 ms, minimum = 10 ms, maximum = 200 ms) to generate an approximately flat hazard function. The total time of the stimulus was usually 1000 ms. Trials of all modalities and stimulus strengths were interleaved. For multisensory trials, the same number of auditory and visual events were presented (except for a subset of neutral trials). Auditory and visual stimulus event times were generated independently, as our previous work has demonstrated that rats make nearly identical decisions regardless of whether stimulus events are presented synchronously or independently (*Raposo et al., 2012*). For most experiments, rats were rewarded with a drop of water for moving to the left reward port following low-rate trials and to the right reward port following high-rate trials. For muscimol inactivation experiments, half of the rats were rewarded according to the reverse contingency. Animals typically completed between 700 and 1200 trials per day. Most experiments had 18 conditions (three modalities, eight stimulus strengths), leading to 29–50 trials per condition per day.

To probe the effect of uncertainty on lapses, rats received catch trials consisting of multisensory neutral trials, where only the auditory modality provided evidence for a particular choice, whereas

the visual modality provided evidence that was so close to the category boundary (12 Hz) that it did not support one choice or the other (*Raposo et al., 2012*).

To probe the effect of value on lapses, we manipulated either reward magnitude or reward probability associated with high rates, while keeping low-rate trials unchanged. To increase or decrease reward magnitude associated with high rates, the amount of water dispensed on the right port was increased or decreased to 36 µL or 16 µL respectively, while the reward on the left port was maintained at 24 µL. To manipulate reward probability, we occasionally rewarded rats on the (incorrect) left port on high-rate trials with a probability of 0.5. The right port was still rewarded with a probability of 1 on high rates, and reward probabilities on low-rate trials were unchanged (one on the left port, 0 on the right).

## Analysis of behavioral data

### Psychometric curves

Descriptive four-parameter psychometric functions were fit to choice data using the Palamedes toolbox (*Prins and Kingdom, 2018*). Psychometric functions were parameterized as:

$$\psi(x; \mu, \sigma, \gamma, \lambda) = \phi(x; \mu, \sigma)(1 - \lambda - \gamma) + \gamma \tag{1}$$

where $\gamma$ and $\lambda$ are the lower and upper asymptote of the psychometric function, which parametrize the lapse rates on low and high rates, respectively. $\phi$ is a cumulative normal function; $x$ is the event rate, that is, the number of flashes or beeps presented during the 1 s stimulus period; $\mu$ parametrizes the $x$-value at the midpoint of the psychometric function and $\sigma$ describes the inverse slope. 95% Confidence intervals on these parameters were generated via bootstrapping based on 1000 simulations.

Our definition of lapses is restricted to strictly *asymptotic* errors following *Wichmann and Hill, 2001*, and not simply errors on the easiest stimuli tested. Errors on the easiest stimuli could in general arise not just from lapses (strictly defined) but also from perceptual errors caused by low sensitivity to the stimulus, an insufficient stimulus range or non-stationary weights (*Busse et al., 2011*; *Roy et al., 2018*). However, we do not consider easy errors alone to be evidence of lapses and only consider asymptotic errors. To confirm the necessity of including the lapse parameters, we fit the following variants of the model above, including lapse parameters when warranted by model comparison using AIC/BIC:

### No lapses

This model forces $\lambda = \gamma = 0$ for all conditions (visual, auditory, and multisensory) and only allows $\sigma$ and $\mu$ parameters to vary across conditions.

### Fixed lapses

This model allows for a fixed $\lambda$ and $\gamma$ (which may be unequal) across all conditions.

### Restricted lapses

This model allows $\lambda$ and $\gamma$ to vary across conditions, but restricts $\lambda + \gamma$ to be less than 0.1. This corresponds to an often used prior over total lapse rates, embodying the belief that lapse trials are infrequent (*Wichmann and Hill, 2001*; *Prins and Kingdom, 2018*).

### Variable lapses

This model allows both $\lambda$ and $\gamma$ to vary freely across conditions, allowing them each to take any value between 0 and 1 (as long as their sum also lies between 0 and 1).

## Modeling

### Ideal observer model

We can specify an ideal observer model for our task using Bayesian Decision Theory (*Dayan and Daw, 2008*). This observer maintains probability distributions over previously experienced stimuli and choices, computes the posterior probability of each action being correct given its observations, and picks the action that yields the highest expected reward.

Let the true category on any given trial be $c_{true}$, the true stimulus rate be $s_{true}$, and the animal's noisy visual and auditory observations of $s_{true}$ be $x_V$ and $x_A$, respectively. We assume that the two sensory channels are corrupted by independent Gaussian noise with standard deviation $\sigma_A$ and $\sigma_V$, respectively, giving rise to conditionally independent observations.

$$
\begin{aligned}
p(x_A|s_{true}) &= \mathcal{N}(s_{true}, \sigma_A), \quad p(x_V|s_{true}) = \mathcal{N}(s_{true}, \sigma_V), \\
p(x_A, x_V|s_{true}) &= p(x_A|s_{true})p(x_V|s_{true})
\end{aligned}
\tag{2}
$$

The ideal observer can use this knowledge to compute the likelihood of seeing the current trial's observations as a function of the hypothesized stimulus rate $s$. This likelihood $\mathcal{L}$ is a Gaussian function of $s$ with a mean given by a weighted sum of the observations $x_A$ and $x_V$:

$$
\begin{aligned}
\mathcal{L}(s) &= p(x_A, x_V|s) = p(x_A|s)p(x_V|s) \\
&\propto \mathcal{N}(\mu_M, \sigma_M) \\
\mu_M &= w_A x_A + w_V x_V \\
\sigma_M &= (\sigma_A^{-2} + \sigma_V^{-2})^{-\frac{1}{2}} \\
w_A &= \frac{\sigma_M^2}{\sigma_A^2}, \quad w_V = \frac{\sigma_M^2}{\sigma_V^2}
\end{aligned}
\tag{3}
$$

The likelihood of seeing the observations as a function of the hypothesized category $c$ is given by marginalizing over all possible hypothesized stimulus rates. Let the experimentally imposed category boundary be $\mu_0$, such that stimulus rates are considered high when $s > \mu_0$ and low when $s < \mu_0$. Then,

$$
\begin{aligned}
\mathcal{L}(c = \text{High}) &= p(x_A, x_V|c = \text{High}) \\
&= \int_s p(x_A, x_V, s|c = \text{High})ds \\
&= \int_s p(x_A, x_V|s)p(s|c = \text{High})ds \quad \because x_A, x_V \perp c|s \\
&= \int_{s > \mu_0} p(x_A, x_V|s)ds \\
&\propto 1 - \Phi(\mu_0; \mu_M, \sigma_M)
\end{aligned}
\tag{4}
$$

where $\Phi$ is the cumulative normal function. Using Bayes' rule, the ideal observer can then compute the probability that the current trial was high or low rate given the observations, that is, the posterior probability.

$$
\begin{aligned}
p(c|x_A, x_V) &= \frac{p(x_A, x_V|c)p(c)}{p(x_A, x_V)} \\
\Longrightarrow p(c = \text{High}|x_A, x_V) &\propto p_{High}(1 - \Phi(\mu_0; \mu_M, \sigma_M)) \\
\Longrightarrow p(c = \text{Low}|x_A, x_V) &\propto p_{Low}\Phi(\mu_0; \mu_M, \sigma_M)
\end{aligned}
\tag{5}
$$

where $p_{High}$ and $p_{Low}$ are the prior probabilities of high and low rates respectively. The expected value $Q(a)$ of choosing right or left actions (also known as the action values) is obtained by marginalizing the learnt value of state-action pairs $q(c, a)$ over the unobserved state $c$.

$$
\begin{aligned}
Q(a = R) &= p(\text{High}|x_A, x_V)q(High, R) + p(\text{Low}|x_A, x_V)q(Low, R) \\
Q(a = L) &= p(\text{High}|x_A, x_V)q(High, L) + p(\text{Low}|x_A, x_V)q(Low, L)
\end{aligned}
\tag{6}
$$

Under the standard contingency, high rates are rewarded on the right and low rates on the left, so for a trained observer that has fully learnt the contingency, $q(High, R) \to r_R, q(High, L) \to 0, q(Low, R) \to 0, q(Low, L) \to r_L$, with $r_R$ and $r_L$ being reward magnitudes for rightward and leftward actions. This simplifies the action values to:

$$
\begin{aligned}
Q(R) &= p(\text{High}|x_A, x_V)r_R \propto p_{High}(1 - \Phi(\mu_0; \mu_M, \sigma_M))r_R \\
Q(L) &= p(\text{Low}|x_A, x_V)r_L \propto p_{Low}\Phi(\mu_0; \mu_M, \sigma_M)r_L
\end{aligned}
\tag{7}
$$

The max-reward decision rule involves picking the action $\hat{a}$ with the highest expected reward:

$$\hat{a} = \text{argmax}\ Q(a)$$
$$\text{i.e.}\, \hat{a} = R \Longleftrightarrow Q(R) \quad > Q(L)$$
$$\Longleftrightarrow p_{High}(1 - \Phi(\mu_0; \mu_M, \sigma_M))r_R \quad > p_{Low}\Phi(\mu_0; \mu_M, \sigma_M)r_L$$
$$\Longleftrightarrow \Phi(\mu_M; \mu_0, \sigma_M) \quad > \frac{1}{1 + \frac{p_{High}r_R}{p_{Low}r_L}}$$
$$\Longleftrightarrow w_A x_A + w_V x_V \quad > \Phi^{-1}\left(\frac{1}{1 + \frac{p_{High}r_R}{p_{Low}r_L}}; \mu_0, (\sigma_A^{-2} + \sigma_V^{-2})^{-\frac{1}{2}}\right)$$

(8)

In the special case of equal rewards and uniform stimulus and category priors, this reduces to choosing right when the weighted sum of observations is to the right of the true category boundary, that is, $w_A x_A + w_V x_V > \mu_0$. Note that this is a deterministic decision rule for any given observations $x_A$ and $x_V$; however, since these are noisy and Gaussian distributed around the true stimulus rate $s_{true}$, the likelihood of making a rightward decision is given by the cumulative Gaussian function $\Phi$:

$$\text{For}\quad p_{High} = p_{Low}, r_R = r_L$$
$$p(\hat{a} = R|s) = p(w_A x_A + w_V x_V > \mu_0|s)$$
$$= \Phi(s_{true}; \mu_0, \sigma)$$
$$\sigma = \begin{cases} \sigma_A \text{ on auditory trials} \\ \sigma_V \text{ on visual trials} \\ (\sigma_A^{-2} + \sigma_V^{-2})^{\frac{1}{2}} \text{ on multisensory trials} \end{cases}$$

(9)

We can measure this probability empirically through the psychometric curve. Fitting it with a two-parameter cumulative Gaussian function yields $\mu$ and $\sigma$ which can be compared to ideal observer predictions. The $\sigma$ parameter is then taken to reflect sensory noise; and with the assumption of uniform priors and equal rewards, the $\mu$ parameter is taken to reflect the subjective category boundary. For the purpose of assessing optimality of integration, $\sigma$ was individually fit to each condition and compared to ideal observer predictions, but for the purpose of comparing theoretical models of lapses, $\sigma$ on multisensory conditions was constrained to be optimal for all models. Although $\mu$ should equal $\mu_0$ for the ideal observer, in practice it is treated as a free parameter in all models, and deviations of $\mu$ from $\mu_0$ could reflect any of three possible suboptimalities: (1) a subjective category boundary mismatched to the true one, possibly arising from the use of irrelevant features such as total event count (*Odoemene et al., 2018*), (2) mismatched priors, or (3) unequal subjective rewards $r_R$ and $r_L$ of the two actions.

## Inattention model

The traditional model for lapse rates assumes that on a fixed proportion of trials, the animal fails to pay attention to the stimulus, guessing randomly between the two actions. We can incorporate this suboptimality into the ideal observer above as follows: Let the probability of attending be $p_{attend}$. Then, on $1 - p_{attend}$ fraction of trials, the animal does not attend to the stimulus (i.e. receives no evidence), effectively making $\sigma_{sensory} \to \infty$ and giving rise to a posterior that is equal to the prior. On these trials, the animal may choose to maximize this prior (always picking the option that is more likely a priori, guessing with 50–50 probability if both options are equally likely), or probability-match the prior (guessing in proportion to its prior). Let us call this guessing probability $p_{bias}$. Then, the probability of a rightward decision is given by marginalizing over the attentional state:

$$p(\hat{a} = R|s) = p(\hat{a} = R|s, \text{attend})p(\text{attend}) + p(\hat{a} = R|s, \sim\text{attend})p(\sim\text{attend})$$
$$= p(\hat{a} = R|s)p_{attend} + p_{bias}(1 - p_{attend})$$

(10)

Comparing this with the standard four-parameter sigmoid used in psychometric fitting, we obtain

$$p(\hat{a} = R|s_{true}) = \gamma + (1 - \gamma - \lambda)\Phi(s_{true}; \mu_0, \sigma)$$
$$\gamma + \lambda = 1 - p_{attend}, \quad \frac{\gamma}{\gamma + \lambda} = p_{bias}$$

(11)

where $\gamma$ and $\lambda$ are the lower and upper asymptotes respectively, collectively known as 'lapses'. In this model, the sum of the two lapses depends on the probability of attending, which could be

modulated in a bottom-up fashion by the salience of the stimulus; their ratio depends on the guessing probability, which in turn depends on the observer's priors and subjective rewards $r_R$ and $r_L$.

## Motor error/$\epsilon$ greedy model

Lapses can also occur if the observer does not always pick the reward-maximizing or 'exploit' decision. This might occur due to random errors in motor execution on a small fraction of trials given by ε, or it might reflect a deliberate propensity to occasionally make random 'exploratory' choices to gather information about rules and rewards. This is known as an ε-greedy decision rule, where the observer chooses randomly (or according to $p_{bias}$) on ε fraction of trials. Both these models yield predictions similar to those of the inattention model:

$$
\begin{aligned}
p(\hat{a} = R|s) &= p(\hat{a} = R|s)(1-\epsilon) + \epsilon p_{bias} \\
\gamma + \lambda &= \epsilon, \quad \frac{\gamma}{\gamma + \lambda} = p_{bias}
\end{aligned}
\tag{12}
$$

## Uncertainty guided exploration model

A more sophisticated form of exploration is the 'softmax' decision rule, which explores options in proportion to their expected rewards, allowing for a balance between exploration and exploitation through the tuning of a parameter β known as inverse temperature. In particular, in conditions of greater uncertainty about rules or rewards, it is advantageous to be more exploratory and have a lower β. This form of uncertainty-guided exploration is known as Thompson sampling. It can be implemented by sampling from a belief distribution over expected rewards and maximizing with respect to the sample, reducing to a softmax rule whose $\beta$ depends on the total uncertainty in expected reward (*Gershman, 2018*).

$$
\begin{aligned}
p(\hat{a} = R|Q(a)) &= \frac{\exp \beta Q(R)}{\exp \beta Q(L) + \exp \beta Q(R)} \\
&= \frac{1}{1 + \exp(-\beta(Q(R) - Q(L)))}
\end{aligned}
\tag{13}
$$

The proportion of rightward choices conditioned on the true stimulus rate is then obtained by marginalizing over the latent action values $Q(a)$, using the fact that the choice depends on $s$ only through its effect on $Q(a)$, where $\rho$ is the animal's posterior belief in a high-rate stimulus, that is, $\rho = p(c = High|x_A, x_V)$. $\rho$ is often referred to as the *belief state* in reinforcement learning problems involving partial observability such as our task.

$$
\begin{aligned}
p(\hat{a} = R|s) &= \int_{Q(a)} p(\hat{a} = R, Q(a)|s)dQ \\
&= \int_{Q(a)} p(\hat{a} = R|Q(a))p(Q(a)|s)dQ \quad \because \hat{a} \perp s|Q(a) \\
&= \int_{\rho} \frac{1}{1 + \exp -\beta(\rho(r_R + r_L) - r_L)} \frac{\mathcal{N}(\Phi^{-1}(1-\rho, 0, \sigma_{post}), \mu_0 - s, \sigma_{post})}{\mathcal{N}(\Phi^{-1}(1-\rho, 0, \sigma_{post}), 0, \sigma_{post})} d\rho
\end{aligned}
\tag{14}
$$

Since lapses are the asymptotic probabilities of the lesser rewarding action at extremely easy stimulus rates, we can derive them from this expression by setting $\rho \to 1$ or $\rho \to 0$. This yields

$$
\gamma = \frac{1}{1 + \exp(\beta r_L)}, \quad \lambda = \frac{1}{1 + \exp(\beta r_R)}
\tag{15}
$$

Critically, in this model, the upper and lower lapses are dissociable, depending only on the rightward or leftward rewards, respectively. In practice since β can only be specified up to an arbitrary scaling of reward magnitudes, we either fix $r_L$=1 and fit β and a reward bias $\frac{r_R}{r_L}$ in units of $r_L$ (for conditions with different expected β), or fix $\beta = 1$ and fit $r_L$ and $r_R$ in units of $\beta$ (for conditions with the same β where one of the rewards is expected to change).

Such a softmax decision rule has been used to account for suboptimalities in value-based decisions (*Dayan and Daw, 2008*); however, it has not been used to account for lapses in perceptual decisions. Other suboptimal decision rules described in perceptual decisions, such as generalized probability matching or posterior sampling (*Acerbi et al., 2014*; *Drugowitsch et al., 2016*;

*Ortega and Braun, 2013*), amount to a softmax on log-posteriors or log-expected values, rather than on expected values, and do not produce lapses since in these decision rules, when the posterior probability goes to 1, so does the decision probability.

$$p(\hat{a} = R|Q(a)) = \frac{\exp \beta \log Q_R}{\exp \beta \log Q_L + \exp \beta_R} = \frac{Q_R^\beta}{Q_L^\beta + Q_R^\beta} \Rightarrow \begin{cases} \rho \to 1 \Rightarrow p(R) \to 1 \\ \rho \to 0 \Rightarrow p(R) \to 0 \end{cases} \tag{16}$$

### Inactivation modeling
Inactivations were modeled using the following one-parameter perturbations to the decision-making process, while keeping all other parameters fixed:

### Biased evidence
A fixed amount of evidence was added to all modalities. This corresponds to adding a rate bias of $K * \sigma_i$ for a condition with sensory noise $\sigma_i$ with $K > 0$ fixed across modalities, leading to bigger biases for conditions with higher sensory noise.

### Biased value
The expected values of one of the actions was scaled down by a fixed factor of $K < 1$ across all modalities. For instance, $Q_{Li} \to K * Q_{Li}$ produced a rightward biased value for a condition with baseline leftward expected value $Q_{Li}$. This led to a stimulus-dependent bias in action value and consequently lapses, since $Q_{Li}$ is large and heavily affected for low-rate trials, and close to zero and largely unaffected for high-rate trials.

### Biased effort
A fixed 'effort' cost (i.e. negative value) $K < 0$ was added to the expected values of one of the actions for all modalities. This added a stimulus-independent bias in action values, since the difference in expected values was biased away from the effortful action by the same amount irrespective of the stimulus rate.

## Model fitting
Model fits were obtained from custom maximum likelihood fitting code using MATLAB's fmincon, by maximizing the marginal likelihood of rightward choices given the stimulus on each trial as computed from each model. Confidence intervals for fit parameters were generated using the hessian obtained from fmincon. Fits to multiple conditions were performed jointly, taking into account any linear or nonlinear (e.g. optimality) constraints on parameters across conditions. Model comparisons were done using AIC and BIC. For comparisons of fits to data pooled across subjects, AIC/BIC values were computed with respect to the best fit model, so that the best model had an AIC/BIC of 0. For comparisons of fits to individual subject data, AIC/BIC values for each subject were computed with respect to the best fit model for each subject, so that the best model for that subject had an AIC/BIC of 0, and then summed across subjects.

## Surgical procedures
All rats subject to surgery were anesthetized with 1–3% isoflurane. Isoflurane anesthesia was maintained by monitoring respiration, heart rate, oxygen, and $CO_2$ levels, as well as foot pinch responses throughout the surgical procedure. Ophthalmic ointment was applied to keep the eyes moistened throughout surgery. After scalp shaving, the skin was cleaned with 70% ethanol and 5% betadine solution. Lidocaine solution was injected below the scalp to provide local analgesia prior to performing scalp incisions. Meloxicam (5 mg/mL) was administered subcutaneously (2 mg/kg) for analgesia at the beginning of the surgery, and daily 2–3 days post-surgery. The animals were allowed at least 7 days to recover before behavioral training.

### Viral injections
Two rats, 15 weeks of age, were anesthetized and placed in a stereotaxic apparatus (Kopf Instruments). Small craniotomies were made in the center of primary visual cortex (V1; 6.9 mm posterior to Bregma, 4.2 mm to the right of midline) and primary auditory cortex (A1; 4.7 mm posterior to Bregma, 7 mm to the right of midline). Small durotomies were performed at each craniotomy and

virus was pressure injected at depths of 600, 800, and 1000 µm below the pia (150 nL/depth). Virus injections were performed using Drummond Nanoject III, which enables automated delivery of small volumes of virus. To minimize virus spread, the Nanoject was programmed to inject slowly: fifteen 10 nL boluses, 30 s apart. Each bolus was delivered at 10 nL/s. Two to three minutes were allowed following injection at each depth to allow for diffusion of virus. The AAV2.CB7.CI.EGFP.WPRE.RBG construct was injected in V1, and the AAV2.CAG.tdTomato.WPRE.SV40 construct was injected in A1. Viruses were obtained from the University of Pennsylvania vector core.

### Cannulae implants

Rats were anesthetized and placed in the stereotax as described above. After incision and skull cleaning, two skull screws were implanted to add more surface area for the dental cement. For striatal implants, two craniotomies were made, one each side of the skull (3.2 mm posterior to Bregma; 5.4 mm to the right and left of midline). Durotomies were performed and a guide cannula (22 gauge, 8.5 mm long; PlasticsOne) was placed in the brain, 4.1 mm below the pia at each craniotomy. For secondary motor cortex implants, one large craniotomy spanning the right and left M2 was performed (~5 mm × ~2 mm in size centered around 2 mm anterior to Bregma and 3.1 mm to the right and left of midline). A durotomy was performed and a double guide cannula (22 gauge, 4 mm long; PlasticsOne) was placed in the brain, 300 µm below the pia. The exposed brain was covered with sterile Vaseline and cannulae were anchored to the skull with dental acrylic (Relyx). Single or double dummy cannulae protruding 0.7 mm below the guide cannulae were inserted.

## Inactivation with muscimol

Rats were lightly anesthetized with isoflurane. Muscimol was unilaterally infused into pStr or M2 with a final concentration of 0.075–0.125 µg and 0.1–0.5 µg, respectively. A single/double-internal cannula (PlasticsOne), connected to a 2 µL syringe (Hamilton microliter syringe, 7000 series), was inserted into each previously implanted guide cannula. Internal cannulae protruded 0.5 mm below the guide. Muscimol was delivered using an infusion pump (Harvard PHD 22/2000) at a rate of 0.1 µL/min. Internal cannulae were kept in the brain for three additional minutes to allow for diffusion of muscimol. Rats were removed from anesthesia and returned to cages for 15 min before beginning behavioral sessions. The same procedure was used in control sessions, where muscimol was replaced with sterile saline.

## Histology

At the conclusion of inactivation experiments, animals were deeply anesthetized with Euthasol (pentobarbital and phenytoin). Animals were perfused transcardially with 4% paraformaldehyde. Brains were extracted and post-fixed in 4% paraformaldehyde for 24–48 hr. After post-fixing, 50–100 µm coronal sections were cut on a vibratome (Leica) and imaged.

## Acknowledgements

We thank Matt Kaufman, Simon Musall, Onyekachi Odoemene, Ashley Juavinett, Farzaneh Najafi, Akihiro Funamizu, Priyanka Gupta, Anne Urai, James Roach, Colin Stoneking, Diksha Gupta, Tatiana Engel, Rob Phillips, Tony Zador, Steve Shea, and Bo Li for scientific advice and discussions, and Angela Licata, Steven Gluf, Liete Einchorn, Dennis Maharjan, Alexa Pagliaro, Edward Lu, and Barry Burbach for technical assistance. We thank Partha Mitra, Alexander Tolpygo, and Stephen Savoia for help with slicing and imaging virus-injected brains. This work was supported by the Simons Collaboration on the Global Brain, ONR MURI, the Eleanor Schwartz Fund, the Pew Charitable Trust, and the Watson School of Biological Sciences. [Competing Interests] The authors declare that they have no competing financial interests. [Correspondence] Correspondence and requests for materials should be addressed to Anne K Churchland (email: churchland@cshl.edu).

## Additional information

### Funding

| Funder | Grant reference number | Author |
|---|---|---|
| Army Research Office | W911NF-16-1-0368 | Sashank Pisupati<br>Lital Chartarifsky-Lynn<br>Anup Khanal<br>Anne K Churchland |
| National Institutes of Health | R01 EY022979 | Anne K Churchland |

The funders had no role in study design, data collection and interpretation, or the decision to submit the work for publication.

### Author contributions

Sashank Pisupati, Data curation, Formal analysis, Validation, Investigation, Visualization, Writing - original draft, Writing - review and editing; Lital Chartarifsky-Lynn, Formal analysis, Investigation, Visualization, Writing - review and editing; Anup Khanal, Investigation, Writing - review and editing; Anne K Churchland, Conceptualization, Resources, Data curation, Supervision, Writing - original draft, Project administration, Writing - review and editing

### Author ORCIDs

Sashank Pisupati https://orcid.org/0000-0003-0923-0585
Lital Chartarifsky-Lynn https://orcid.org/0000-0003-2446-9842
Anup Khanal https://orcid.org/0000-0002-8929-7984
Anne K Churchland https://orcid.org/0000-0002-3205-3794

### Ethics

Animal experimentation: All animal procedures and experiments were in accordance with the National Institutes of Healths Guide for the Care and Use of Laboratory Animals and were approved by the Cold Spring Harbor Laboratory Animal Care and Use Committee (protocol 19-16-13-10-7).

### Decision letter and Author response

Decision letter https://doi.org/10.7554/eLife.55490.sa1
Author response https://doi.org/10.7554/eLife.55490.sa2

## Additional files

### Supplementary files

• Transparent reporting form

### Data availability

Data are publicly available: http://repository.cshl.edu/id/eprint/38957/.

The following dataset was generated:

| Author(s) | Year | Dataset title | Dataset URL | Database and Identifier |
|---|---|---|---|---|
| Pisupati S, Chartarifsky L, Khanal A, Churchland A K | 2020 | Dataset from: Lapses in perceptual decisions reflect exploration | https://doi.org/10.14224/1.38957 | CSHL, 10.14224/1.38957 |

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
