## [Decision Letter]

**Acceptance summary:**

This manuscript presents a novel explanation for lapses in perceptual decision making. Using precise computational models to analyze the data from a click-rate discrimination task, the authors show that lapses might be due to the rats' uncertainty-dependent exploration strategy rather than inattention or motor errors.

**Decision letter after peer review:**

Thank you for submitting your article "Lapses in perceptual decisions reflect exploration" for consideration by *eLife*. Your article has been reviewed by three peer reviewers, and the evaluation has been overseen by a Reviewing Editor and Joshua Gold as the Senior Editor. The following individuals involved in review of your submission have agreed to reveal their identity: Long Ding (Reviewer #1); Alex C Kwan (Reviewer #2).

The reviewers have discussed the reviews with one another and the Reviewing Editor has drafted this decision to help you prepare a revised submission.

Summary:

This manuscript presents an interesting explanation for lapses in perceptual decision making. The authors showed that rats showed imperfect performance on a click-rate-discrimination task. They showed that the amount of lapse depended on whether the decision was based on uni- or multi-sensory stimulus, whether the reward associations were equal or asymmetric for the two choices, and whether M2/pStr was intact. Using precise computational models, they proposed that these lapses were due to the rats' uncertainty-dependent exploration strategy. The results are somewhat consistent with the uncertainty-dependent exploration strategy, but some features that are inconsistent with this strategy appear to be ignored. A critical analysis to directly relate uncertainty and lapse across sessions was also missing. Therefore, although the manuscript has the potential of establishing the uncertainty-dependent exploration strategy as one of the factors contributing to lapses, additional analyses/explanations are needed.

Essential revisions:

1) The contrast between regular multisensory trials and "neutral" trials is a clever design. However, the results did not convincingly support the uncertainty-dependent exploration model. As the authors stated "the lapse parameter on neutral trials should match those on auditory trials, since these conditions have comparable levels of perceptual uncertainty". Judging from Figure 3—figure supplement 1E, this prediction did not hold for the example rat. It is unclear how well it held for the other four rats.

2) Model comparison results in Figure 3—figure supplement 3C suggest that the inattention model performed as well as the Exploration model for fitting uni/multi-sensory data. However, the results of the model fitting should be more fully disclosed. The results in Figure 3—figure supplement 3G were not conclusive. Namely, the inattention model seemed to outperform the Exploration model for rat#4; both models performed similarly for rat #5, and the Exploration model was better for the other three rats. The example rat in Figure 3—figure supplement 1E also showed other behavioral patterns that are puzzling. When reward was increased for the Right choice, there appeared to be a leftward bias (comparing the "Multisensory" curve in the left panel and the "Increased Right" curve in the right panel). The "equal reward" curve in the right panel showed significantly worse performance than other curves. How representative were these behavioral patterns? Do these patterns invalidate the uncertainty-dependent exploration model?

3) The two models (inattention and fixed error) used to compare against the exploration model are simplistic and may not serve as a fair comparison. In particular, based on prior literature on similar rodent task, it seems that another model based on motivation + inattention might be a more relevant and reasonable explanation, and should be compared against the exploration model. There is evidence that in sensory discrimination tasks, rodent's behavior exhibits serial choice bias. Specifically, if the last trial yielded a reward, then that could influence the current decision (Busse et al., 2011; Siniscalchi et al., 2019). One reasonable interpretation is that this is a motivational component that is dependent on the prior trial's outcome. Given this, one model that may be worthwhile to try is an outcome-dependent inattention model, where the amount of inattention differs depending on whether the last trial was rewarded or not. Namely, if the last trial was rewarded, then animal has fewer lapses, whereas if the last trial was not rewarded, then animal has more lapses. There is indication that some aspects of the current data support this idea (Figure 4F). How would this type of model contrast with the exploration model? One specific question is, similar to Figure 4F, but if we additionally plot previous L success and previous L failure, then does the reward history for prior L choices influence the proportion of choosing R at high stimulus rate?

The premise is that the exploratory choices would resemble lapses. This is true in a task design involving two choice options, but probably should be considered as a caveat of the task design. If the task has more than two choices, then one may more confidently distinguish these processes and identify periods of exploration. Some considerations as to how such a task design (or the fact that the current finding only has two options) influences the conclusions should be added in the Discussion.

4) The claim is that there is uncertainty-driven exploration that could explain the lapse rate. However, the task always employs the same criterion boundary for the discrimination problem, and the stimulus set is fixed across sessions. The animals are presumably over-trained and expert in this task, so it is unclear why they would be incentivized to update values for the stimuli in this sensory discrimination task. The authors presented some data to suggest they continuously learn. Is there a normative explanation for why they should be doing this in the current experiments?

5) Although the data in Figure 4C appear to support the uncertainty-dependent exploration model, it is possible that, on equal reward trials, the three rats trained for the "increased r_Right_" condition performed much worse than the three rats trained for the "decreased r_Right_" condition. The difference in "Proportion choose high" at 16Hz between the two cohorts for equal reward trials appeared as large as the effects of changing reward. The differences between equal reward trials and "increased/decreased r_Right_" trials might be due to some factors beyond value associations (e.g., how the two cohorts were trained).

6) There are many variants of models in the manuscript, but they were not presented in sufficient details, making it hard to track what parameters were fixed or fitted separately for different types of trials in a given experiment. For example, for the data in Figure 5, the legend says that the model fits scaled all contralateral values by a single parameter. Does it mean that this scaler was the only free parameter for the inactivation data, after fitting the control data? Or the model was fitted to both control and inactivation data simultaneously, with all but the scaler fixed between the two datasets? If a single scaling parameter can account for the inactivation effects, similar effects would be expected for auditory, visual and multi-sensory decisions for a given rat. But this does not seem to be the case. For example, Rats 8,9,10 in Figure 5—figure supplement 3 showed very different effects between auditory and visual decisions for M2-low rate side inactivation. Similarly, rats 2,3,6 in Figure 9—figure supplement 4 for pStr-low rate side inactivation. It would be helpful to have a table with the fitted parameter values for each experiment/rat, so that readers can better track how the model fitting was done and develop a better sense of how changes in model parameters affect the psychometric curves.

[Editors' note: further revisions were suggested prior to acceptance, as described below.]

Thank you for submitting your article "Lapses in perceptual decisions reflect exploration" for consideration by *eLife*. Your article has been reviewed by three peer reviewers, and the evaluation has been overseen by a Reviewing Editor and Joshua Gold as the Senior Editor. The following individuals involved in review of your submission have agreed to reveal their identity: Long Ding (Reviewer #1); Alex C Kwan (Reviewer #2).

The reviewers have discussed the reviews with one another and the Reviewing Editor has drafted this decision to help you prepare a revised submission.

Summary:

This manuscript presents an interesting explanation for lapses in perceptual decision making. The authors showed that rats showed imperfect performance on a click-rate-discrimination task. They showed that the amount of lapse depended on whether the decision was based on uni- or multi-sensory stimulus, whether the reward associations were equal or asymmetric for the two choices, and whether M2/pStr was intact. Using precise computational models, they proposed that these lapses were due to the rats' uncertainty-dependent exploration strategy. The authors have addressed most of the concerns raised by the reviewers appropriately, but there is one issue that requires additional clarification.

The original figure of concern actually showed example neutral/auditory trials from different rats. The authors generated new figures showing both types of trials from 5 rats separately (Figure 3—figure supplement 1E and Author response image 2C). In three out of the five rats, for the LOW choice, the lapse was larger for neutral trials; for the HIGH choice, the lapse was larger for auditory trials. This kind of asymmetric difference in lapse appears similar to the predictions for effort manipulation in Figure 4—figure supplement 2. If there was no category-specific value/effort manipulation between neutral and auditory trials, it is not intuitive how the uncertainty-dependent exploration model can account for this asymmetry. An explanation would be helpful.

---

## [Author Response]

Essential revisions:1) The contrast between regular multisensory trials and "neutral" trials is a clever design. However, the results did not convincingly support the uncertainty-dependent exploration model. As the authors stated "the lapse parameter on neutral trials should match those on auditory trials, since these conditions have comparable levels of perceptual uncertainty". Judging from Figure 3—figure supplement 1E, this prediction did not hold for the example rat. It is unclear how well it held for the other four rats.

We thank the reviewers for these kind words about the experimental design and we are very grateful that they brought this issue to our attention. The example rats in Figure 3—figure supplement 1E were intended to demonstrate the relationship between conditions *within* each manipulation, hence a different example rat was chosen for each manipulation (Author response image 1 shows the original figure). This yielded misleading patterns *across* manipulations, especially since these were incorrectly labelled "example rat" in the figure and legend. We thank the reviewer for bringing these discrepancies to our attention – we have corrected this and chosen the same example rat across all manipulations (same rat, lc40, as reward panel – Author response image 1) in order to facilitate across-manipulation comparisons. We hope that this revised version of the figure makes it clear that the total lapses on neutral trials (Author response image 1, middle panel, orange trace) are similar to those on auditory trials (Author response image 1, left panel, green trace).

In addition:(1) An unconstrained descriptive model ( "variable lapse" i.e. 4 independent parameters per condition) fit to all 5 rats reveals that slope and lapse parameters on neutral and auditory conditions lie along the unity line for 4/5 rats (Author response image 2) We have updated the Results section to reflect this.

(2) The exploration model fits (Author response image 2) demonstrate that the prediction *did* hold for the example rat (1st panel) as well as the other 4 rats: auditory and neutral conditions were constrained to have the same σ and lapse parameters in the exploration model (Author response image 2), and this provides a good fit to the data.

**Author response image 2. respfig2:** 

2) Model comparison results in Figure 3—figure supplement 3C suggest that the inattention model performed as well as the Exploration model for fitting uni/multi-sensory data. However, the results of the model fitting should be more fully disclosed. The results in Figure 3—figure supplement 3G were not conclusive. Namely, the inattention model seemed to outperform the Exploration model for rat#4; both models performed similarly for rat #5, and the Exploration model was better for the other three rats. The example rat in Figure 3—figure supplement 1E also showed other behavioral patterns that are puzzling. When reward was increased for the Right choice, there appeared to be a leftward bias (comparing the "Multisensory" curve in the left panel and the "Increased Right" curve in the right panel). The "equal reward" curve in the right panel showed significantly worse performance than other curves. How representative were these behavioral patterns? Do these patterns invalidate the uncertainty-dependent exploration model?

The reviewer is correct in pointing out that there is individual variability in the best fitting model in Figure 3—figure supplement 3; however it is important to note that the ideal observer model is a limiting case of the inattention/exploration models, and for animals with very small lapse rates, these models are indistinguishable so BIC would strongly prefer the more parsimonious ideal observer model  - this is the case for both rats 4 and 5 for which the ideal observer provides the best fit according to BIC, suggesting that these rats have very small lapse rates. All 3 rats that are rejected by the ideal observer model i.e. have sizable lapse rates, are best fit by the exploration model. In the revised version of the manuscript, we acknowledge this variability in individuals that are best fit by the ideal observer vs exploration models (Results).

As for the rat in (Figure 3—figure supplement 1E), the "equal reward" and "increased Right" conditions on the rightmost panel both consist of auditory trials, and the performance on “equal reward” trials is comparable to performance on auditory trials in the “multisensory” or “neutral” experiments (updated left, center panel from the same example rat). We have clarified this in the legend.

3) The two models (inattention and fixed error) used to compare against the exploration model are simplistic and may not serve as a fair comparison. In particular, based on prior literature on similar rodent task, it seems that another model based on motivation + inattention might be a more relevant and reasonable explanation, and should be compared against the exploration model. There is evidence that in sensory discrimination tasks, rodent's behavior exhibits serial choice bias. Specifically, if the last trial yielded a reward, then that could influence the current decision (Busse et al., 2011; Siniscalchi et al., 2019). One reasonable interpretation is that this is a motivational component that is dependent on the prior trial's outcome. Given this, one model that may be worthwhile to try is an outcome-dependent inattention model, where the amount of inattention differs depending on whether the last trial was rewarded or not. Namely, if the last trial was rewarded, then animal has fewer lapses, whereas if the last trial was not rewarded, then animal has more lapses. There is indication that some aspects of the current data support this idea (Figure 4F). How would this type of model contrast with the exploration model? One specific question is, similar to Figure 4F, but if we additionally plot previous L success and previous L failure, then does the reward history for prior L choices influence the proportion of choosing R at high stimulus rate?The premise is that the exploratory choices would resemble lapses. This is true in a task design involving two choice options, but probably should be considered as a caveat of the task design. If the task has more than two choices, then one may more confidently distinguish these processes and identify periods of exploration. Some considerations as to how such a task design (or the fact that the current finding only has two options) influences the conclusions should be added in the Discussion.

We fully agree with the reviewers here. If the level of attention were changed following correct/error trials, this could potentially account for the trial history effects: an increase/decrease in attention following success/failure, in combination with an upward or downward shift due to changes in guessing probabilities emerging from updates to the prior/action values (i.e. following success/failure on the R, animal assumes that High/Low rates are more likely -or- assumes that R actions are more/less rewarding.)

Importantly, in order for this to capture the asymmetry in the data, the two effects would have to be fine-tuned to cancel each other out at low rates and add up at high rates. Further, since this model invokes trial-by-trial updates, a fair comparison to it would be an exploration model combined with trial-by-trial updates of the values of chosen actions based on past outcomes (thus producing the asymmetry, which is seen following both leftward and rightward actions – Author response image 3), which is how we currently propose outcomes affect subsequent trials, as we allude to in the discussion on trial-by-trial modeling.

**Author response image 3. respfig3:** 

However, we can still make predictions for trial-averaged behavior from this model for the different manipulations, by assuming that p(Attend) is modulated by average reward i.e. higher average rewards give rise to greater overall motivation, more attention and fewer lapses. This allows us to compare its predictions with other models on the neutral and reward manipulations:(1) Predictions for matched v. neutral: Since the “matched” and “neutral” trial types are randomized and uncued, the animal doesn't know what the upcoming condition is and the average rewards preceding the two trials should be the same. Hence, the motivated inattention model predicts the same level of attention across the two conditions, just like the regular inattention model, predicting equal lapses (unlike those observed in the data)

(2) Predictions for reward magnitude manipulation: The motivated inattention model can indeed explain the effects of the reward magnitude experiment, by assuming that the higher/lower average reward on increased/decreased reward conditions gives rise to more/less attention, and fewer/more lapses in conjunction with the upward/downward shifts predicted by the regular inattention model. Once again, this does require fine-tuning of the two effects in order to cancel out at low rates. (Author response image 4 left, center panels)

3) Predictions for reward probability experiment: In this experiment, leftward actions are probabilistically rewarded (50%) on *highrates* (instead of yielding 0 reward), and always rewarded on low rates – thus increasing the overall proportion of rewarded outcomes and increasing the proportion of leftward trials rewarded, compared to rightward trials.

– This predicts an increase in overall attention (to both stimulus categories) due to the higher levels of motivation, and hence a decrease in overall lapses

– It also predicts a non-specific increase in leftward choices due to the leftward biased average rewards yielding more leftward inattentive guesses.

In particular, these two effects should lead to a bigger downward shift in low rates (as shown in Author response image 4 right panel)

However, this is not the effect observed in the data, instead the manipulation only increases leftward choices *for high rates*, thus *increasing* lapses on high rates, and consequently increasing the overall lapse rate – which matches the predictions of the exploration model.

**Author response image 4. respfig4:** 

We thank the reviewers for the motivated inattention model, and have added it as a new panel (Figure 4—figure supplement 1F). We have also added text in the Discussion about the caveats of a task design with two options, and cited emerging work that offers a possible remedy (Zatka-Haas et al., 2018, Mihali et al., 2018).

4) The claim is that there is uncertainty-driven exploration that could explain the lapse rate. However, the task always employs the same criterion boundary for the discrimination problem, and the stimulus set is fixed across sessions. The animals are presumably over-trained and expert in this task, so it is unclear why they would be incentivized to update values for the stimuli in this sensory discrimination task. The authors presented some data to suggest they continuously learn. Is there a normative explanation for why they should be doing this in the current experiments?

The reviewers are correct that in the current task, the true category boundary, stimulus-action contingency and expected rewards are fixed. Therefore the normative strategy is to explore until the uncertainty in action values reduces to zero, and then stop exploring. While some features of the task, such as sensory uncertainty, abstractness of the stimulus-response contingency and arbitrariness of the category boundary could lead to increased action value uncertainty to begin with, this normative strategy still predicts zero lapses in the asymptotic limit of training (e.g. Author response image 5 simulation with belief in stationary rewards, increasing sensory uncertainty – indicated by cooler colors – reduces speed of reduction in lapse rates, but asymptotic lapse rate is 0).

A possible normative explanation for the fact that the animals are continuously learning/exploring is that they do *not* assume that the action values are static, but instead entertain the possibility that they drift/change over time (possibly reflecting the statistics of real world rewards). While this model is clearly mismatched to the current task, under this model (i.e. in truly non-stationary worlds) the normative solution is to maintain a low level of exploration to test whether the world has changed or not – we can simulate this using a model that assumes a small rate of non-stationarity in values (Author response image 5.) This model predicts a residual level of uncertainty that never goes to 0, and consequently a residual asymptotic lapse rate scaled by sensory uncertainty. Some animals do indeed achieve close-to-zero lapse rates after extensive training (e.g. Figure 4—figure supplement 2), but for animals that still have residual lapses, it is difficult to distinguish these models with finite training data.

Fortunately, a unique prediction of a mismatched world model is that it predicts that in the event of a contingency change, subjects should unlearn old contingencies much faster than predicted by a stationary belief model (which would display perseveration and take as many examples to unlearn a contingency as it did to learn it – akin to Ebitz et al., 2019). We have performed preliminary tests of this prediction by reversing the contingency in a small cohort of rats. We observed that rats did indeed unlearn old contingencies much faster than predicted from the stationary belief model, and resembled a model that entertains a small possibility of non-stationarity in values.

**Author response image 5. respfig5:** 

We have expanded on this point about normativity and included predictions for matched/mismatched world models and tests of contingency change in the Discussion. However, we have not included the results of our contingency change experiments in this manuscript since they are quite preliminary.

5) Although the data in Figure 4C appear to support the uncertainty-dependent exploration model, it is possible that, on equal reward trials, the three rats trained for the "increased r_Right_" condition performed much worse than the three rats trained for the "decreased r_Right_" condition. The difference in "Proportion choose high" at 16Hz between the two cohorts for equal reward trials appeared as large as the effects of changing reward. The differences between equal reward trials and "increased/decreased r_Right_" trials might be due to some factors beyond value associations (e.g., how the two cohorts were trained).

The reviewer is correct in pointing out that there are individual differences between cohorts at baseline, possibly due to differences between the precise history of training data each rat has seen (which we expect to reflect in the subjective action values learnt by each rat).

For this reason, we restrict all our reward comparisons in Figure 4C to within-cohort comparisons – each rat is first trained on equal reward, then tested with a reward manipulation (either increase or decrease) to measure the effect of the reward manipulation on its behavior relative to baseline (which in the exploration model is captured by scaling only the relevant baseline value). Neither of the cohorts have baseline lapses at ceiling/floor, allowing for measurement of the effect of reward manipulation on both lapses and hence a comparison of the different models. Moreover, the exploration model captures within-individual comparisons relative to baseline by changing high rate action values on the right alone, even though individuals vary substantially in their baseline left/right values (Table 2). We have clarified this in the Results.

6) There are many variants of models in the manuscript, but they were not presented in sufficient details, making it hard to track what parameters were fixed or fitted separately for different types of trials in a given experiment. For example, for the data in Figure 5, the legend says that the model fits scaled all contralateral values by a single parameter. Does it mean that this scaler was the only free parameter for the inactivation data, after fitting the control data? Or the model was fitted to both control and inactivation data simultaneously, with all but the scaler fixed between the two datasets? If a single scaling parameter can account for the inactivation effects, similar effects would be expected for auditory, visual and multi-sensory decisions for a given rat. But this does not seem to be the case. For example, Rats 8,9,10 in Figure 5—figure supplement 3 showed very different effects between auditory and visual decisions for M2-low rate side inactivation. Similarly, rats 2,3,6 in Figure 5—figure supplement 4 for pStr-low rate side inactivation. It would be helpful to have a table with the fitted parameter values for each experiment/rat, so that readers can better track how the model fitting was done and develop a better sense of how changes in model parameters affect the psychometric curves.

We agree. First, to address the point about clarifying models and parameters in the paper, we generated a new table with fit parameters for all the experiments and models in order to clarify parameters and constraints for each of the fits. (Table 1 i.e. Figure 4—source data 1: fits to pooled data across individuals. We generated a second table, Table 2, with individual fits. However, we did not include this in the revised manuscript because we feared this might be cumbersome to include in the supplement).

As for inactivation fits, the model was fit to both control and inactivation data simultaneously, with a single scalar being the only parameter differing between the two datasets. We have added a separate section describing inactivation modeling in the Materials and methods to clarify this point.

Second, we have addressed the issue that the fits in (Figure 5—figure supplement 3,4) were unclear. In the original manuscript, they were indeed all descriptive, unconstrained fits (i.e. 4 parameters per fit x 3 modalities x 2 perturbation conditions = 24 params per rat). We have updated this figure with individual fits of the best fitting model for each rat (biased evidence/value/effort, which have 11 control + scalar = 12 params per rat). Despite being heavily constrained, these account for all rats quite well.

All three models of inactivation are capable of producing effects with differing strengths across modalities: the key intuition is that these effects interact with the baseline sensory noise (biased evidence) and baseline values/exploratoriness (biased value and biased effort), producing the strongest effects for modalities with the highest sensory noise/exploration. The simulation (Author response image 6) illustrates this difference. Performance on control trials for left- and right-biased baseline action valus shows only subtle differences (compare solid traces on top, bottom). However, the same inactivation strength (i.e. multiplicative/additive factor) drives strikingly different effects on these conditions (compare dashed traces for top, bottom). This highlights the strength of a model that can estimate action values from lapse rates. This approach can reconcile seemingly different effects across conditions within the same animal (or across animals) with a single change in action value, without needing to invoke separate mechanisms for each condition.

**Author response image 6. respfig6:** 

However despite the ability of the model to account for disparate changes in the psychometric function, we agree with the reviewer's assessment that the biased value model does not fully account for the effects in some rats. In Rats 8,9 (M2 low rate), 2(pStr High),3(pStr low) and 6(pStr high and low), the best fitting model was still one in which action values changed. But, importantly, the winning model for those rats was the one in which a single scalar 'effort' (i.e. negative, stimulus-independent value) was added to the contralateral side, rather than a single multiplicative (i.e. stimulus-dependent) value scaling. This suggests that the inactivations affected value additively, rather than multiplicatively in these rats. In the revised version of the text, we address these individual differences (and display fits from the best fitting model for each rat in Figure 5—figure supplements 3,4), and acknowledge that additional recording and inactivation studies might shed light on why disruptions drive changes that are sometimes additive and sometimes multiplicative. We argue that our approach nonetheless gives an experimenter considerable power in interpreting inactivations that diverge across conditions.[Editors' note: further revisions were suggested prior to acceptance, as described below.]

Revisions for this paper:The original figure of concern actually showed example neutral/auditory trials from different rats. The authors generated new figures showing both types of trials from 5 rats separately (Figure 3—figure supplement 1E and Author response image 2). In three out of the five rats, for the LOW choice, the lapse was larger for neutral trials; for the HIGH choice, the lapse was larger for auditory trials. This kind of asymmetric difference in lapse appears similar to the predictions for effort manipulation in Figure 4—figure supplement 2. If there was no category-specific value/effort manipulation between neutral and auditory trials, it is not intuitive how the uncertainty-dependent exploration model can account for this asymmetry. An explanation would be helpful.

The reviewers have correctly identified that while auditory and neutral trials have comparable sigmas and total lapse rates, some rats indeed show a slight low-rate bias on auditory trials compared to neutral and multisensory trials (Author response image 8). While all models considered can account for this through their free “bias” and “lapse-bias” parameters (hence not affecting model comparisons), none of them explain its existence.

Instead, we think that this bias can be explained by animals not using a pure "flash/pulse rate" decoder to solve the task, but instead using a hybrid decoder that incorporates both "flash/pulse rate" and "total flash/pulse count" information. These two features are correlated with each other and with reward on all trial types, making them both susceptible to reward-based credit assignment – we reported a similar hybrid "rate+count" strategy in rats and mice on a visual-only variant of the rate task (Odoemene et al., 2018, Figure 3). Note that neutral and multisensory trials however are offset on the "count" dimension compared to auditory trials, since they have additional events for the same "rate" (twice as many on multisensory, 12 additional on neutral – First column in Author response image 7).

As a result, only an observer that uses a pure "rate" decoder would have an unbiased psychometric across all conditions, centered at the true category boundary of 12.5 (First row in Author response image 7) – this is true of rat 5 in Author response image 8. However, even a weak influence of count would bias conditions with respect to each other. For instance, an animal that uses a "hybrid" decoder that is unbiased for multisensory and neutral conditions would be biased towards low rates on auditory trials, with increasing use of count information producing greater biases (Second, third, fourth rows of Author response image 7 – effects of count information adding 1/15th as much, 1/7th as much or just as much evidence as rate information).

Such a hybrid decoder would translate into different *effective* state-action value pairs for different conditions based on their position in this 2-d space, with unisensory conditions having the highest asymmetry between low and high rate action values. In the fixed error model, this asymmetry would simply produce horizontal biases in the psychometric function, but in both inattention and exploration models, this asymmetry would additionally bias the lapses. Hence, the varying biases and lapse biases seen in the other 4 rats (Author response image 8) could arise from varying degrees of use of count information. Note that if an animal using a hybrid decoder was unbiased on auditory trials, then it would show a high-rate bias on multisensory trials, as is observed in some of the animals in the 1st cohort (unisensory vs. multisensory)

We tested this “hybrid decoder” hypothesis in an independent cohort of rats where we increased/decreased the duration of auditory trials to increase/decrease count information without changing rate information (as per Odoemene et al., 2018), and indeed found evidence for a hybrid decoder with a weak influence of count (Author response image 9), however we think these results are outside the scope of the current manuscript. In case other readers share the reviewers’ concern about bias, we now mention the count bias and refer to previous work.

**Author response image 7. respfig7:** Use of count information generates bias in psychometric functions. Simulations demonstrating the effect of 4 different linear decoders (1st column) on psychometric functions under the fixed error (2nd column), inattention (3rd column) and exploration (4th column) models. Dotted black lines indicate the true category boundary that separates “low” and “high” stimulus categories, solid black lines indicate the subjective category boundary for each decoder. Colored dots indicate stimulus sets in auditory (green), neutral (orange) and multisensory (red) conditions. Open circles indicate point of subjective equality i.e. stimulus that produces equal “high” and “low” evidence for each condition. Rate-only decoder (1st row) aligns with the true category boundary, producing unbiased psychometric functions across conditions, with models differing only in the total lapse rate of each condition. Hybrid decoders incorporating a small amount of count information (2nd row: 1/15th as much as rate, 3rd row: 1/5th as much as rate, 4th row: just as much as rate) that are unbiased on multisensory, neutral conditions produce a low-rate bias on auditory trials, with more use of count information producing higher bias. This produces horizontal shifts (i.e. bias) across models, and additionally produces vertical shifts (i.e. lapse bias) in inattention, exploration models, while preserving the predictions for total lapse rates across models/conditions.

**Author response image 8. respfig8:** Bias in psychometric functions of individual rats. Data from the 5 rats used in the neutral experiment, on auditory (green), neutral (orange) and multisensory (red) conditions, demonstrating individual variability in low-rate bias on auditory trials, compared to neutral/multisensory trials. Some rats (e.g. rightmost panel) show almost no bias, resembling the “rate-only” decoders, while others resemble “hybrid” decoders with weak influences fromcount information.

**Author response image 9. respfig9:** Duration manipulation on auditory trials confirms weak influence of count. Data from 4 rats (right) shows that increasing (light green) or decreasing (brown) the duration of auditory trials produces slight high- or low-rate biases, resembling a hybrid decoder with a weak influence of count information(middle panel left).